# Peptide-Based Hydrogels: New Materials for Biosensing and Biomedical Applications

**DOI:** 10.3390/ma15175871

**Published:** 2022-08-25

**Authors:** Roya Binaymotlagh, Laura Chronopoulou, Farid Hajareh Haghighi, Ilaria Fratoddi, Cleofe Palocci

**Affiliations:** 1Department of Chemistry, Sapienza University of Rome, Piazzale Aldo Moro 5, 00185 Rome, Italy; 2Research Center for Applied Sciences to the Safeguard of Environment and Cultural Heritage (CIABC), Sapienza University of Rome, Piazzale Aldo Moro 5, 00185 Rome, Italy

**Keywords:** peptide-based hydrogels, antibacterial properties, drug delivery, biosensing, tissue engineering, vaccines

## Abstract

Peptide-based hydrogels have attracted increasing attention for biological applications and diagnostic research due to their impressive features including biocompatibility and biodegradability, injectability, mechanical stability, high water absorption capacity, and tissue-like elasticity. The aim of this review will be to present an updated report on the advancement of peptide-based hydrogels research activity in recent years in the field of anticancer drug delivery, antimicrobial and wound healing materials, 3D bioprinting and tissue engineering, and vaccines. Additionally, the biosensing applications of this key group of hydrogels will be discussed mainly focusing the attention on cancer detection.

## 1. Introduction

Hydrogels are constituted by a three-dimensional (3D) network of insoluble materials that swell by absorbing large amounts of water [1]. They are promising candidates for biomedical applications due to their high water-absorbing capacity, chemical and mechanical stability and tunable microporous structures, good biocompatibility, and biodegradability. More importantly, there are standardized methods to functionalize hydrogels with a variety of biologically important species, allowing the design of multi-functional platforms to fabricate advanced hybrid materials [2]. Hydrogels can be classified as synthetic or natural, based on the type of their building block polymers. Peptide hydrogels are a sub-group of natural hydrogels that are extremely attractive for biomedical applications [3]. The cost of peptide synthesis is comparable to some polymeric hydrogels, but in terms of tuning peptide synthesis, reaction conditions and purification steps, peptide hydrogels are superior to synthetic hydrogels [4]. From a synthetic point of view, these hydrogels can be prepared by the physical assembly of peptides through noncovalent interactions, including hydrogen bonding, electrostatic, hydrophobic, and *π*–*π* interactions. On the other side, they can also be fabricated by the chemical and enzymatic cross-linking of peptides (see Table 1). Depending on the specific biological application, these hydrogels can be used (1) by conjugation/combination with other biologically active components or (2) alone, where hydrogels themselves act as bioactive drugs, vaccines, biosensors, or regulating factors for physiological functions in vivo or in vitro [1,5].

In peptide hydrogels, amino acids are the building blocks of a 3D matrix and, depending on the number of peptide linkages (–CONH– amide linkage), we can differentiate between short and long chain peptide hydrogels. In this regard, dipeptides to oligopeptides are classified as short chain peptides, which are cost effective and have easier synthesis methods compared to long chain ones. Obviously, the physiochemical properties of the formed hydrogels directly depend on the type of the peptides employed, and so amino acids and their sequence in the peptide chains are the two main important parameters that affect the final properties of the hydrogels; for instance, the nature of amino acids can determine the hydrophobicity and hydrophilicity, thus affecting the gelation properties. Additionally, increasing the number of peptide bonds in the main chain, the intermolecular hydrogen bonding between the peptide chain increases, affecting the physical properties of the hydrogels, such as mechanical strength. As a practical point of view, for example, the drug loading and release behaviors of the peptide hydrogels strongly depend on the types of the amino acids and their sequence [13].

Being composed of the smallest short peptides, dipeptide hydrogels are a popular group of hydrogels due to their low cost and well-known synthesis methods; thus, these hydrogels are commonly studied for different biological applications (see Table 2). From structural investigations, it is well known that this type of hydrogels requires the presence of additional aromatic groups in their structures to enhance gel formation and mechanical strength through the self-assembly process [14,15,16]. For instance, L-Phe-L-Phe (FF) peptide is a popular building block for hydrogel synthesis; however, FF cannot self-assemble into supramolecular networks without the presence of additional aromatic groups such as fluorenylmethyloxycarbonyl (Fmoc) or longer peptides containing the FF motif [17,18].

In this regard, Guilbaud-Chéreau et al. studied the self-assembly of aromatic dipeptides (Boc-*α*-diphenylalanine and Boc-*α*-dityrosine) and their *β*- and *γ*-homologues to form stable hydrogels [19]. Interestingly, only the Boc-diphenylalanine analogues self-assembled into stable hydrogels. Then the authors encapsulated L-ascorbic acid, and oxidized carbon nanotubes (or graphene oxide) inside the hydrogels and studied the release of L-ascorbic acid through light-assisted photothermal effect of carbon nanomaterials, resulting in a high amount of drug release. Shim et al. combined 9-fluorenylmethoxycarbonyl-modified diphenylalanine (Fmoc-FF) with glycol chitosan (GCS) (by electrostatic interaction) to fabricate a novel chitosan-dipeptide hydrogel [20]. Then they incorporated doxorubicin (DOX) as a cationic model drug into the hydrogel, which showed slow DOX release and excellent thixotropic properties. Zou et al. synthesized a novel amphiphile bola-dipeptide (a di-FF derivative, DFF, containing aromatic groups) and the related fibrous hydrogels in aqueous solution for the delivery of a hydrophilic phototherapeutic drug, 5-aminolevolinic acid [21]. They reported that this hydrogel showed good stability, rapid recovery, and excellent biocompatibility. In another work, Ren et al. synthesized several Fmoc-functionalized dipeptides, including phenylalanine-leucine (FL), tyrosine-leucine (YL), leucine-leucine (LL), and tyrosine-alanine (YA), and studied their self-assembly behaviors, tuning their mechanical strength by altering the amino acid sequence to modify hydrophobic and hydrophilic interactions among the molecules [22]. Gavel et al. developed a class of biocompatible Amoc (9-anthracenemethoxycarbonyl)-capped dipeptide hydrogels exhibiting injectability and self-healing properties [23,24]. They used suitable aromatic groups in the gelator molecules, demonstrating excellent gelation ability due to their ability to form *π*–*π* stacking and hydrophobic interactions.

Dipeptides can also be used in combination with enzymes (as biosensors) and these hydrogel-enzyme hybrids are used in smart biointerfaces. For example, Lian et al. prepared Fmoc-diphenylalanine nanofibrous hydrogels combined with horseradish peroxidase (HRP), during the self-assembly of Fmoc-FF peptides, and used them in biosensing and cell monitoring applications [25]. In another study, they employed dipeptides for immobilizing live cells and enzymes as sensing elements into the hydrogel matrix for the detection of oxygen ion (O_2_^•−^) [59]. Baral et al. synthesized a modified dipeptide hydrogel by self-assembly of an N-Boc protected dipeptide with the assistance of the Phe (L-phenylalanine) moiety present in the gelator through *π*–*π* stacking and hydrophobic interactions [26]. Vilaça et al. reported dehydrodipeptide hydrogels containing an anti-inflammatory drug, naproxen, formed by hydrogen-bonding and *π*–*π* stacking interactions [27]. From fluorescence studies, they deduced that this hydrophobic drug was located in a hydrophobic environment of the hydrogel, indicating the potential ability of the hydrogel to increase the therapeutic efficacy of hydrophobic drugs.

Tripeptides are the next class of short peptides that can be N-terminal protected [60] or non-protected [13]. In tripeptides, the contribution of aromatic groups or other *π*-conjugated systems in the peptide structure plays a key role for gel formation through *π*–*π*-stacking interactions. For example, Yeh et al. synthesized naphthalimide capped tripeptide hydrogelators (NI-GFF (NI-Gly-L-Phe-L-Phe) and NI-FFG (NI-L-Phe-L-Phe-Gly)) and studied their structural properties correlation [28]. With a small variation in the peptide chain in NI-GFF and NI-FFG, the authors observed a large difference between their self-assembly properties. Under the same conditions and at neutral pH, NI-GFF self-assembled into a stable hydrogel, whereas NI-FFG did not. Das Hernández-Sosa et al. used different alginate–cellulose formulations for employing them as bioinks in 3D hydrogel-based scaffolds [29]. In these structures, alginate (Alg) acts as the main component, because of its fast gelation and tunable mechanical properties, and cellulose modulates the mechanical properties of the printing. Regarding tripeptide hydrogels, Fmoc-FFY (Fmoc: 9-fluorenylmethoxycarbonyl; F: phenylalanine; Y: tyrosine) promoted cell adhesion to the resulting scaffold without changing its rheological properties. Recently, Li et al. employed a tripeptide-based macroporous alginate hydrogel to enhance the osteogenic differentiation efficiency of stem cells [30]. An arginine-glycine-aspartate peptide hydrogel improved the adhesion and growth of stem cells, and the in vivo results showed the formation of more blood vessels inside the hydrogel compared to other systems, demonstrating a high potential of such hydrogel systems for tissue engineering applications. Yang et al. reported that the Fmoc-protected phenylalanine-phenylalanine-cysteine could self-assemble into micelles at high pH values, followed by a spontaneous transformation into a stable hydrogel [31]. Mahapatra et al. prepared a series of pH-sensitive Fmoc-modified tripeptide hydrogels containing *β*-amino acid showing thermoreversibility and proteolytic stability (against proteinase K enzyme) [32]. In our group, there have been several studies on the synthesis, characterizations, and applications of modified tripeptides (Fmoc-FFF), which showed high self-assembling properties, by using lipases [33,34,35,36,37,38,39]. The high number of aromatic groups present in Fmoc-FFF peptides enhances the self-assembly process through *π*–*π* stacking and hydrogen bonding to form hydrogel networks. In one study, we investigated the lipase-catalyzed formation of self-assembling tripeptides made of all D-amino acids or alternating DLL and LDD amino acids [36,61]. The results showed that, in addition to the formation of nanofibers, the D-form and mixed DL peptides were able to self-organize into 3D nanofiber scaffolds, like their L-counterpart. The peptides can form a *β*-sheet-like molecular structure with D-chirality to give the mirror image of the circular dichroism (CD) spectrum of the L-peptide counterpart. In terms of swelling ability of the hydrogel (an important index of its interaction capability with the aqueous based environment in the biological site of application), we reported that all hydrogels (either L or D) present very similar and high values of swelling ratios [36,61].

As another category of short peptide hydrogels, tetrapeptides can also self-assemble into 3D nanostructured hydrogels through non-covalent interactions [62,63]. Recently, Chen et al. reported a simple method to prepare H7N9 vaccines using tetrapeptide adjuvant (GDFDFDY) hydrogels [40]. Since 2013, the influenza virus H7N9 has caused severe influenza in humans, with mortality rates of about 40–50%. As a good anti-inflammatory drug, naproxen, was employed as the hydrogel precursor to form the product Npx-GDFDFDY. Their results showed the best preventive effects of this hydrogel against H7N9 infection, compared with the traditional Al(OH)_3_ adjuvant vaccine, exhibiting the potential ability of such tetrapeptide hydrogels in vaccines against other viruses. Alshehri et al. prepared two different tetrapeptide hydrogels consisting of IVFK and IVZK sequences, specifically composed of a positively charged amino acid (Lys) in the C-terminal and three nonpolar amino acids as a hydrophobic tail [41]. These two tetrapeptides were self-assembled into ordered aggregates due to their amphiphilic nature. Dewangan et al. reported the synthesis of phenolic acid-modified tetrapeptide injectable hydrogels and showed the dominant role of *π*–*π* interactions in self-assembly and hydrogelation [42]. They studied the entrapment and sustained release of two antibiotics, rifampicin and ciprofloxacin, at physiological conditions for 5 days. The antibacterial studies against clinically relevant multidrug resistant *S. aureus* (MRSA), showed a moderate antibacterial activity of the native hydrogel and a significant synergistic effect upon the entrapment of antibiotics. More importantly, this hydrogel exhibited a biocompatibility on hRBCs and human keratinocytes up to hydrogelation concentrations. Zhao et al. used a tetrapeptide hydrogel for the encapsulation of protein molecules to form a hydrogel-vaccine system [43]. Compared to the native hydrogel, the mechanical properties of this hydrogel-vaccine system are enhanced up to 15-fold with a conformational change from *β*-sheet to *α*-helix structure. Wu et al. employed healing–cooling protocols to synthesize a naproxen acid modified tetrapeptide (Npx-GFFY) hydrogel using both D and L amino acids [44,45,63]. Liang et al. loaded a commercial eye drop, diclofenac sodium (DIC), into Nap-protected tetrapeptide hydrogels for ophthalmic drug delivery applications [46]. They reported the rapid release of this drug within 24 h without any toxicity (against different cell lines) or eye irritation.

Pentapeptide hydrogels are of great importance in terms of synthesis and biological applications [47]. Chen et al. reported a family of pentapeptide hydrogels and studied the thermodynamics, gelation kinetics, and ionization status of terminal groups of the pentapeptides, reporting that the structures of the hydrogels are highly dependent on the charging status of the terminal groups [48]. Moreover, they encapsulated a hydrophobic functional molecule (curcumin) inside the nanostructures with high efficiency and significant protection of the drug against thermal and light degradation. In 2021, Yin et al. synthesized a three-dimensional pentapeptide IKVAV-functionalized poly (lactide ethylene oxide fumarate) (PLEOF) hydrogel [49]. This biodegradable and hemocompatible hydrogel showed neural stem cell attachment, growth, proliferation, and differentiation. Tang et al. synthesized a new family of pentapeptide hydrogels and showed the dependance of mechanical properties of these hydrogels with the sequence variations of the used peptides, pH, and peptide concentration [50]. Lys amino acid was used as the head group of the peptides to modify the hydrophilicity of the hydrogel; however the overall sequence of the pentapeptides was designed specifically to increase the amphiphilicity of the peptides. In another study, they developed an injectable pentapeptide hydrogel used as a scaffold that exhibited shear-thinning and rapid self-healing properties [51]. Due to having a nanofiber morphology, this hydrogel can serve as a cell carrier in transplantation applications and tissue engineering. Clarke et al. synthesized a pentapeptide hydrogel comprising only two amino acids, isoleucine and aspartate, in the peptide sequence [47]. They showed the change of gelification behavior with altering the concentration and charged distribution.

The self-assembly in hexapeptide hydrogels provides ease of gelation and mechanical stability. Luo et al. synthesized a hexapeptide hydrogel having high pH sensitivity, which is degraded in the acidic microenvironment, for the delivery and sustained release of doxorubicin (DOX) [53]. Their results indicated that this hydrogel-DOX system has a good pH-responsivity and injectability with high therapeutic effect and less toxicity on normal tissues. In 2021, Diaferia et al. synthesized six analogous of amphiphilic cationic peptides and studied the effect of the addition of aromatic groups (such as Fmoc) in their structures to explore the gelation properties of the resulting hydrogels [52]. In these derivatives, the hydrogelation is facilitated by additional *π*–*π* stacking due to the presence of the aromatic moiety. Their results showed that those hydrogels having both the Fmoc group and the FF motif lose the key interactions for the self-assembly process, while those with only an Fmoc group keep their capability for hydrogel formation. Such reports show the importance of peptide sequence and the type of amino acids on the chemical features and self-assembly of the resulting hydrogels. In these hydrogels, due to the stacking of the aromatic group, peptides form *β*-sheet structures with an antiparallel orientation of the *β*-strands. Mei et al. synthesized a fluorescent hexapeptide hydrogel comprising a fluorescent moiety for the accurate and reversible measurement of pH [54]. Qi et al. developed a hexapeptide hydrogel with nanofiber morphology, which interestingly is changed into well-organized nanofibers having antiparallel *β*-sheets in the presence of DOX [55]. Their in vitro results showed that the local injection of this hydrogel-DOX system in mice inhibits the re-growth of cancer cells and reduces distant metastasis.

Oligopeptides are considered short chain peptides made by connecting more than three and less than twenty-one peptides, and they have gained a great interest for a variety of biotechnological applications. Like the previously mentioned hydrogels, oligopeptides can also form hydrogels through non-covalent self-assembly, resulting in a variety of nanostructures (including *β*-sheets, helical, or *β*-hairpin-based fibrils and nanotubes) [13]. Li et al. synthesized a novel injectable octapeptide hydrogel (formed at pH 7.4) as a drug delivery platform for DOX [56]. In another research, they reported a heparin-containing oligopeptide hydrogel for the co-delivery of a neutralizing antibody (anti-TNF-a) and hepatocyte growth factor (HGF) to expedite tissue repair after ischemia-reperfusion injury [57]. The presence of heparin in the structure of this hydrogel enhances its affinity toward HFG through a molecular shape recognition mechanism. As an important *β*-heparin oligopeptide, Max8 contains twenty amino acids, and it is considered as a stimuli-responsive and stable oligopeptide at low ionic strengths that can self-assemble into *β*-hairpin nanofibers under physiological conditions. Such *β*-hairpin peptide hydrogels are considered drug delivery systems due to their biocompatibility [13]. In 2020, Schneible et al. employed a modified Max8 hydrogel for the co-delivery of chemotherapeutic drugs (DOX and gemcitabine) to breast cancer cells [58]. To control the release kinetics of these two drugs, they used graphene oxide nanoparticles inside the hydrogel.

Other worth mentioning oligopeptides are cyclic peptides, which can be formed by amide linkage between one end of a peptide with another peptide, or even other chemically stable bonds such as lactone, ether, thioether, and disulfide. In more detail, N-to-C (or head-to-tail) cyclization is an amide bond formation between amino and carboxyl termini, which can be found in many biologically active cyclic peptides. Interestingly, cyclic peptide derivatives can form hydrogel structures in aqueous and nonaqueous media through non-covalent self-assembly by physical stimuli such as temperature, shear force, and pH, as reported by Manchineella et al. Such biologically active peptides are found in nature and already used in clinic, such as, for instance, in gramicidin and tyrocidine (bactericidal activity), cyclosporin A (immunosuppressive activity), and vancomycin (antibacterial activity) [64]. Usually, this group of peptides exhibits a more effective biological activity compared to their linear counterparts, due to the conformational rigidity; in spite of this, their hydrogelating properties are largely unexplored [65]. In 1994, Khazanovich et al. reported that cyclic peptides consisting of 8 to 12 amino acids of alternating D- and L-configurations can stack through *β*-sheet antiparallel H-bonding and form well-defined 10–500 nm nanotubes having cavity diameters of 2–13 Å [66]. In 2018, Shaikh et al. reported a class of cyclic peptide gelators [67] forming hydrogels via a sequential hierarchical self-assembly process including (1) vertical stacking of these peptides into hollow nanotubes, (2) aggregation of the nanotubes into larger fibers, and (3) formation of a dense network that can entrap water and form a hydrogel.

In this review paper, we will highlight some representative studies on the biomedical applications of peptide hydrogels, especially in the fields of biosensing, anticancer drug delivery, vaccines, antimicrobial, wound healing materials, 3D bioprinting, and tissue engineering. We hope that this review draws increasing attention to the importance of peptide-based hydrogels and facilitates the advances of hydrogels in biomedical applications, paving the way towards the formulation of advanced biomedical materials.

## 2. Biomedical Applications of Peptide-Based Hydrogels

### 2.1. Biosensing Applications

In recent years, there has been increasing attention towards peptide hydrogel biosensors due to their high responsiveness to external stimuli such as pH and temperature, good cell adhesion, well-known chemistry for structural modification, long-term chemical and mechanical stability, antifouling properties, tunable viscoelastic characteristics, and self-healing features [3,6] (see Figure 1). On the basis of their chemical structure, hydrogels can be divided in two main groups including: (1) peptide hydrogels possessing high environmental sensitivity—for instance to pH, temperature, or electrical fields—which are used alone without bioreceptors or other auxiliary sensing components, and (2) peptide hydrogels having high porous structures and large internal surfaces which are applied in combination with sensing biomolecules, including enzymes, DNA strands, and label molecules. The second type of hydrogel is of great significance because of the prevention of non-specific adsorption on the hydrogel surface, which makes them suitable candidates for the fabrication of ultra-sensitive biosensors [3].

Regarding cancer detection, peptide hydrogels are used to sense biomarkers (biological molecules found in blood or other body fluids discriminating normal and abnormal cells and demonstrating a condition of cancer) due to the large difference between the biomarker concentration in normal cells and cancerous ones [68]. Recently, Zhang et al. fabricated a low-fouling electrochemical sensor demonstrating an accurate detection and capturing of tumor biomarkers in biological solutions. In this system, they used multifunctional peptides and zeolitic imidazolate frameworks (ZIFs) [69]. To prevent nonspecific attachment in biological media, they constructed a hydrophilic interface consisting of sodium alginate-graphene oxide-Pb^2+^ (SA-GO-Pb^2+^) gel and multipotent peptides (containing recognition, antifouling, and anchoring sequences). These multifunctional peptides linked the gel to a novel pyrrole doped ZIF loaded with urease (urease@ZIF-Py). Urease hydrolyzes the added urea generating CO_2_ (g), and this gas can then react with Pb^2+^ to induce PbCO_3_ precipitation, which prevents the electron transfer at the sensing interface, effectively enhancing the sensitivity. For their model studies, they selected matrix metalloproteinase-7 (MMP-7) as a reliable biomarker enzyme for colon cancer detection found in serum. The signals of the sensor were enhanced by increasing MMP-7 concentration. This biosensor platform demonstrated significant antifouling properties in the biological medium used with a desirable linear range (0.1 pg mL^−1^–100 ng mL^−1^) and a low detection limit of 24.34 fg mL^−1^. These results are very promising for developing low-fouling biosensors, which is very useful for complex biological solutions. Huihui et al. used tetrapeptide tryptophan-valine-phenylalanine-tyrosine (WVFY) to construct a high-performance tyrosinase-sensing transistor-based (n-type metal oxide) biosensor (bio-FETs) [70]. This biosensor could rapidly detect the target tyrosinase through the fast conversion of the phenolic hydroxyl groups of the tetrapeptide to benzoquinone, which needs the consumption of protons and is potentiometrically detectable by the bio-FETs. They reported an ultra-low detection limit of 1.9 fM in the optimum range of 10 fM to 1 nM. The biosensors fabricated on solid substrates of ~2.9-μm-thick polyimide (PI) have a high mechanical tunability and, more importantly, they are biocompatible with the human skin. The attachment of such constructs to the skin can lead to the design of wearable melanoma sensors and provide the opportunity for detecting other important cancer biomarkers. Wang et al. employed a short Phe-Glu-Lys-Phe hydrogel for the electrochemical detection of HER2, a breast cancer biomarker, via the chemical attachment of this hydrogel on an antifouling sensing interface made from the conducting polymer poly(3,4-ethylenedioxythiophene) (PEDOT) [71]. For selective HER2 sensing, the authors chemically conjugated anti-HER2 antibody with the hydrogels, providing high sensitivity and selectivity for HER2, detecting concentrations as low as 45 pg mL^−1^ with a linear responsivity. Due to the chemical stability of this biosensor, its workability was tested in human serum, providing clinically acceptable accuracy, manifesting a perspective practical application for cancer detection.

In another study, Lian et al. employed Fmoc-FF hydrogel as a biointerface in an enzyme-based electrochemical sensor for detecting H_2_O_2_ released from HeLa cells [25]. They self-assembled Fmoc-FF peptide on a glassy carbon electrode, followed by the encapsulation of a model enzyme, horseradish peroxidase (HRP), in the hydrogel matrix, which allows retaining the biosensing properties of HRP. Due to the cell adhesive feature of this hydrogel and its biocompatibility, it was tested for the attachment and growing of HeLa cells and in situ sensing of H_2_O_2_ produced by these cells. This biosensor showed a high selectivity and low detection limit (18 nM), which potentially can be developed for sensing other unstable biomarkers. It can be considered as a cost-effective and user-friendly electrochemical biosensor for more studies in the real-time monitoring of various cellular biomolecules.

In another work, an amperometric sensor based on an Fmoc-FF-derived hydrogel for the detection of superoxide anions (O_2_^−^) release from 3D cultured HeLa cancer cells was fabricated [59]. This hydrogel has a dual function including: (1) a 3D cell culture scaffold and (2) a matrix for the immobilization of two enzymes, HRP and superoxide dismutase (SOD). In one step, the two enzymes and HeLa cells were simultaneously embedded inside the Fmoc-FF derived hydrogel, followed by dropping this mixture onto a glassy carbon electrode forming a 49 μm-thick layer of hydrogel–enzyme–cells. Thanks to the biocompatibility and cell adhesion property of the hydrogel, the cells grow and release the superoxide biomarker, which can be detected with a high sensitivity of 85 mA mM^−1^ cm^−2^ and a low detection limit of 0.34 nM.

Regarding virus detection, Gagni et al. employed a YF-Q11 peptide hydrogel for the attachment of biomolecular probes, followed by the deposition of this hydrogel–bioprobe on analytical poly (methyl methacrylate) slides to obtain 3D arrays [72]. The immunodiagnostic performance of this biosensor was studied for the detection of Zika virus (ZIKV) infection in human serum samples. By attaching fluorescent dyes to this matrix and using antibodies, they showed a highly stable and permeable scaffold to run fluorescence immunoassays having ultrashort timescale (<10 min) with a pM detection limit. This hydrogel is printable onto poly (methyl methacrylate) (PMMA) slides, demonstrating a user-friendly and cheap biosensor.

Souza et al. encapsulated *Leishmania infantum chagasi* antigens inside an Fmoc-FF hydrogel to enhance the bioactivity and lifetime of the biomolecules for *Leishmaniasis* detection [73]. Among zoonotic diseases, *Leishmaniasis* is a widespread infection, considered as a serious public health issue that causes relevant mortality and morbidity [15]. The enzyme-linked immunosorbent assay (ELISA) showed the efficiency to sense positive dog sera for *Leishmania infantum chagasi*. In fact, the hydrogel matrix provides a moist and stable biological environment for the biomolecules, and it protects them from degradation over a long time.

King et al. reported the first experiments of a fluorescence-based hydrogel DNA biosensor, which works based on switching between off and on states of the fluorophore in the absence and presence of the target, respectively [74]. To construct this biosensor, they firstly employed the peptide Val-Lys-Val-Lys-Val-Glu-Val-Lys to form flexible fibers comprising antiparallel *β*-sheets stacked perpendicularly to their long axes. Then they chemically conjugated a DNA recognition motif CGATTCTGTGTT to the hydrogel, followed by the hybridization of the DNA motif with a molecular beacon (MB) fluorescent probe F-50-CGATTCGCCA AACACAGAATCG-30-D, where F is fluorescein and D is dabcyl. In the absence of target, the fluorescence of this biosystem is quenched due to the FRET phenomenon; conversely, a strong fluorescence signal is generated when the target DNA is hybridized with the DNA motif and forms a more stable duplex comprising 12 Watson–Crick complementary base pairs, showing a detection limit of 22 pM for DNA detection. This user-friendly biosensor can be developed for the detection of other molecules in medicinal and environmental fields.

Khatayevich et al. used self-assembling multifunctional short peptides (biotin-graphite binding peptide-(bio-GrBP5)) attached on a graphene surface to form a graphene field effect transistor (gFET) biosensor for the selective biosensing of streptavidin without the interference of serum protein (BSA) [75]. For this purpose, the two groups of peptides (probing and non-probing) were engineered and co-assembled on the substrate. The probe peptides detect the streptavidin, and the non-probing ones prevent non-specific BSA adsorption onto the device surface, guaranteeing target selectivity, which can detect the biomolecule at less than 50 ng/mL. With the experimental conditions they used, this biosensor can be restored and reused for 12 experiments. As a perspective view, this biosensor can be developed for sensing biomarkers in clinical samples.

### 2.2. Anticancer Drug Delivery

In traditional cancer chemotherapy, patients often suffer from adverse side effects due to the uncontrolled delivery of drugs to healthy cells, which results in damaging them. To reduce these adverse side effects and enhance therapeutic efficacy, peptide-based hydrogels have been used as drug delivery systems due to their biocompatibility, tunable structure to load different drugs, water-filled mesoporous structures, and sensitivity to external stimuli to induce drug release. In fact, hydrogels often improve the biochemical properties of anticancer drugs in terms of chemical stability, solubility, and bioavailability (see Figure 2). From a chemical point of view, drugs are incorporated within hydrogels by two general modalities, through either chemical or physical interactions. In a recent work, Zhang et al. developed a pH-responsive and injectable peptide hydrogel–MTX delivery system for local and specific tumor chemotherapy, which afforded low side effects [76]. A hydrophilic drug, methotrexate (MTX) was conjugated to the peptide chain (KKFKFEFEF) using a pH-responsive linker, 2,3-dimethylmaleic anhydride, via the formation of amidic bonds. By decreasing the pH from 7.4 to 6.5, the negative charge of this drug-hydrogel system changes to a positive value by a sol-gel phase transition. Both in vitro and in vivo results showed the highly efficient cellular uptake and endocytosis capacities of this system. In fact, the hydrogel could prolong the in vivo anti-tumor therapeutic effect of the loaded MTX with negligible side effects in mice. The results of this study reveal the promising aspects of pH-responsive drug–peptide hydrogels for enhancing the therapeutic efficacy of traditional cancer treatments.

Liu et al. reported the synthesis of an injectable and pH-responsive oligopeptide hydrogel for the co-delivery and controlled release of two therapeutic drugs, gemcitabine (GEM) and paclitaxel (PTX), at the tumor site [77]. They reported the optimal concentration of the peptide for hydrogelation and then studied the in vitro pH-dependent release of this drug delivery platform. At pHs 5.8 and 7.4, PTX showed release amounts of 96.90% and 38.98%, respectively (in 7 days), and GEM exhibited 99.99% (pH 5.8) and 99.63% (pH 7.4) in 3 days. The hydrogels demonstrated pH-dependent structures, had a pH-dependent release character for both drugs, and were able to reach their tumor targets upon injection and showed sustained release kinetics. Li et al. employed an enzymatic approach to fabricate peptide hydrogels using enzyme-coated, carbon-based nanocomposites, which initiate the chemical cross-linking of the peptides and result in the final hydrogels [78]. NIR-photothermal carbon nanotubes (CNTs) were selected and coated with large pore (>10 nm) mesoporous silica (CNT@LPMS) to provide a high enzyme loading. Then, a large amount of enzyme, alkaline phosphatase (AP) (>100% wt%), was entrapped into the porous structure of CNT, which allows the localized self-assembly of peptides, resulting in supramolecular hydrogels. During the hydrogelation process, they loaded DOX and studied its photothermally NIR-induced release. Their experimental and computational studies proved the presence of DOX within the peptide nanofibers. Yang et al. synthesized nanogels containing coiled-coil, self-assembling PC10A peptide for the co-delivery of two common anticancer drugs, PTX and DOX [79]. PTX and DOX contents of the PC10A hydrogel were found to be 5.98 wt% and 70 wt%, respectively, with encapsulation efficiencies of 8.55 wt% (PTX) and 83 wt% (DOX). The in vitro and in vivo antitumor results of this biocompatible hydrogel demonstrated a synergistic therapeutic effect of this system having lower side effects compared with free PTX/DOX. Ghosh et al. employed a new small cyclic dipeptide (CDP) (cyclo-(Leu-S-Bzl-Cys) (P1)) containing **L**-leucine and S-benzyl protected L-cysteine, which could self-assemble into a hydrogel (antiparallel *β*-sheet secondary structure) at 37 °C and pH = 7.46 [80]. The resulting hydrogel exhibited a long-term (for more than a year) structural stability at different pHs (6–12) and temperatures with thermoreversible features. They also studied the hydrogelation of P1 in the presence of the anticancer drug 5-fluorouracil (5FU) and showed the sustainable in vitro release of the drug from the hydrogel. The hydrogelator P1 showed high compatibility with the human colorectal cancer cell line HCT116 until a high concentration, while the combination of 5FU with P1 hydrogel exhibited higher anticancer properties, compared to free 5FU. Liu et al. developed a nano-hybrid oligopeptide hydrogel for the delivery of docetaxel (DTX), as a common chemotherapeutic drug, to prevent post-surgical tumor recurrence [81]. This DTX-CTs/Gel system was fabricated by the entrapment of DTX inside cell-penetrating, peptide-modified transfersomes and the subsequent self-assembly into an oligopeptide hydrogel. The paintable and injectable hydrogels showed prolonged retention at the sites of injection after their administration. The released DTX-CTs demonstrated high skin and tumor penetration and increased the effective DTX concentration in cancer cells, which resulted in an enhanced therapeutic effect, compared to the free drug. More importantly, this work showed that this kind of DTX delivery could efficiently decrease tumor recurrence in post-surgical mouse melanoma and breast tumor models. Guchhait et al. investigated the hydrogelation of three short tripeptides based amphiphiles [11-(2-tert-Butoxycarbonylamino-3-methyl-butyrylamino)-undecanoylamino]-acetic acid TBMBUA (A), [11-(2-tert-Butoxycarbonylamino-3-methyl-pentanoylamino)-undecanoylamino]-acetic acid TBMPUA (B), and [11-(2-tert-Butoxycarbonylamino-3-phenyl-propionylamino)-undecanoylamino]-acetic acid TBPPUA (C) [82]. Their results showed a high hydrogelation ability of the peptides, which were used for loading and releasing Vitamin B12 and DOX. The authors reported a pH-dependent drug release and demonstrated the anticancer activity of the synthesized hydrogelators (TBMBUA, TBMPUA and TBPPUA) on the human breast cancer cell line MCF-7 as well as on the EAC cancer cell line. No toxicity was exhibited towards healthy human lymphocytes up to the concentration of 25 µg/mL. These results demonstrated a great potential of the use of hydrogelators for local drug delivery in breast cancer therapy. Wei et al. used the self-assembling peptide RADA16-I for the in situ synthesis of hydrogels, employed for loading the hydrophobic drug emodin (EM) [83]. This RADA16-I hydrogel showed good cell compatibility and could reduce the toxicity of EM to normal cells and enhance cell uptake, inducing apoptosis, in vitro. In vivo results also confirmed the growth inhibition effects of RADA16-I-EM in mice bearing LLC cells xenograft cancer, initiating cell apoptosis in the cancer tissue, and decreasing the side effects of EM on normal tissues, compared with free EM. Results showed that RADA16-I hydrogel is a promising candidate for the delivery also of other hydrophobic anticancer drugs. Gallo et al. studied DOX loading and release properties of hydrogels and nanogels (both pure and combined) [84]. They also investigated the in vitro cytotoxicity of both hydrogels and nanogels (unloaded and with drug) on MDA-MB-231 breast cancer cell line as Triple Negative Breast Cancer (TNBC), the most aggressive breast cancer subtype. Additionally, they evaluated the encapsulation of supramolecular nanodrugs (for example liposomal DOX formulation, Doxil) into peptide-based hydrogels and their effective cytotoxicity. Their approach provides systems for a multi-stage delivery of DOX. Zou et al. synthesized a novel amphiphilic bola-dipeptide (a di-FF derivative, DFF, containing aromatic groups) and the related fibrous hydrogels for the in vivo delivery of anticancer prodrug, 5-aminolevolinic acid, reporting the advantages of this system, which included: enhanced localized prodrug conversion, inhibited nontargeted leakage, sustained release profiles, and effective tumor ablation [21]. Karavasili et al. studied the possibility of localized co-drug delivery of DOX and curcumin to glioblastoma multiforme, a highly malignant primary brain tumor [85]. They synthesized a biocompatible peptide, ac-(RADA)_4_-CONH_2_, by an in situ approach showing a controlled-release of both hydrophobic (curcumin) and hydrophilic (DOX) chemotherapeutic drugs against U-87 MG cancer cell line. Liu et al. used an RADA16-hydrogel to stabilize PTX [86]. They showed that the RADA16-PTX system could effectively inhibit the growth of the breast cancer cell line MDA-MB-435S in vitro, and hydrogels with higher peptide concentrations were more effective in inhibiting tumor cell proliferation. The RADA16-PTX hydrogel was effective at controlling the release of PTX and inhibiting tumor cell growth in vitro. As a perspective view, such promising studies can result in decreasing the common side effects of PTX in the clinical use.

It is worth mentioning that, because of the inherent biological origins of peptides, peptide hydrogels can act as therapeutic agents for some cancer cell lines. For instance, Chen et al. synthesized a short peptide-based hydrogel with MMP-2 responsiveness (MMP-2 is overexpressed by HeLa cells) for controlled anticancer peptide delivery [87]. They firstly designed and prepared self-assembled fibrillar Ac–I_3_SLGK–NH_2_ hydrogels, which can be degradable in the presence of HeLa cells. They used this feature for the controlled release of an anticancer peptide, G(IIKK)_3_I–NH_2_, in cancer cells.

In topical drug delivery through the skin, there is a high limitation due to the physiological barriers between the skin and the tumor, which can significantly reduce the therapeutic effect. To enhance transdermal efficiency, hydrogels can be used to modify the penetration ability of anticancer drugs across the skin, due to their biocompatibility and tunable structure, allowing the attachment of different modifiers and drugs to hydrogels for this purpose. Jiang et al. attached a cell penetrating peptide, R8H3, modified on transfersomes (formulated by phospholipids and surfactants including Tween 80 and sodium deoxycholate), to PTX [88]. This hydrogel, encapsulating PTX, enhanced skin penetration to the target tumor stroma and showed effective tumor growth inhibition. For drug delivery to superficial tumors, another admirable work conducted by Abbas et al. reported the physical encapsulation of a photosensitizer Ce6 and its controlled delivery in an MCF-7 tumor-bearing mouse model [18]. The hydrogel was formed by physical electrostatic interaction between negatively charged Fmoc-FF and positively charged PLL peptides. By changing the Fmoc-FF/PLL ratios, the authors tuned the mechanical properties of the hydrogel suitable for injection to the target site. Their results showed that this hydrogel can prolong drug concentration and control release at the tumor site, reducing the toxicity of Ce6 through more efficient tumor growth inhibition with less damage to vital organs, compared to the use of free Ce6.

It is worthy to mention that the physical interaction between drug and hydrogel is more common for hydrophilic drugs, which can be easily incorporated into peptide hydrogels. However, the encapsulation of hydrophobic therapeutics often results in very low drug contents, typically in the order of 2–5% by weight and, for this reason, hydrophobic drugs must be loaded into the supramolecular structure. For more than a decade, covalently linked anticancer drugs have been studied, basically using water-soluble peptides. For example, Lin et al. synthesized a glutathione (GSH)-responsive hydrogel chemically attached to a high amount of hydrophobic drug, PTX [89]. In the presence of GSH, PTX is released and effectively inhibits the growth of MCF-7, A549, and PC3-flu cancer cells. In another worth-mentioning work, Gao et al. chemically attached PTX to succinic acid and a self-assembly motif, followed by activation by N-hydroxysuccinimide, to form a highly stable drug delivery system [90]. In the presence of alkaline phosphatase (NapFFKYp), this system transforms into nanofibers to form the final hydrogel-PTX derivative that slowly releases the modified PTX into the aqueous medium. This study opened a way towards the formulation of highly hydrophobic drugs, such as PTX, into hydrophilic systems (e.g., hydrogel) without losing their activity and, more importantly, decreasing their adverse toxicity.

In another paper, the authors studied the chemical conjugation of anti-HIV reverse transcriptase inhibitor, 20,30-dideoxy-30-thiacytidine (3TC) or azidothymidine (AZT), with a peptide hydrogelator to form hydrogels at acidic pH [91]. The biocompatibility and release of the HIV inhibitors under physiological conditions was reported.

Karavasili et al. employed an ac-(RADA)_4_-CONH_2_ hydrogel for the local co-delivery of hydrophilic DOX and hydrophobic curcumin to head and neck cancer cells, in vivo and in vitro [92]. Compared to DOX and curcumin individual drugs, dual drug-loaded hydrogel systems showed a much higher anti-proliferative effect on HSC-3 cells, due to the synergistic effect provided by the hydrogel network. Jin et al. introduced a melittin-RADA_32_ hybrid hydrogel loaded with doxorubicin for effective chemoimmunotherapy against B16-F10 tumor-bearing mice by the activation of immunosuppressive tumor microenvironments [93]. Their results showed that this nanofibrous hydrogel has biocompatibility and controlled-drug release providing a potent immunological memory effect against metastatic tumor cells. Raza et al. synthesized a pH responsive octapeptide hydrogel for the delivery of PTX [94]. The peptide molecules form a stable hydrogel with high drug loading capacity at pH 7.4. The authors reported that this peptide hydrogel transforms into a sol and sustainably releases PTX at the tumor site of mice at pH 5.5, efficiently enhancing tumor inhibition.

Finally, it is worth mentioning that the in vitro studies of peptide-based hydrogels have shown that they can improve the pharmacokinetic parameters of both hydrophilic and hydrophobic drugs and, more importantly, reduce the effective drug dosage required for the anticancer effect. For the clinical approval of such smart drug carriers, more research is needed to strengthen and develop this frontier medicinal topic.

### 2.3. Vaccines

The immune system has a paramount importance for the suppression of immune-related disorders such as cancer. However, in some cases, free antigens have poor immunogenicity and do not have enough effectiveness, and hence, biocompatible immunological adjuvants are employed to improve the antigen immunogenicity [2]. Peptide-based hydrogels are promising candidate for using as immune adjuvant due to their cost-effective large-scale production, biocompatibility, and relatively high stability and activity [95,96,97]. In this regard, there have been excellent publications; for instance, Zhang et al. studied the structural-therapeutic relationships of five synthesized Gel-gffy variants (Sol-gfgy, Sol-ggfy, Gel-gffg, Gel-gfyf, and Gel-gyff) by changing the sequence of amino acids in the parent D-tetrapeptide through substituting the phenylalanine and tyrosine to glycine or changing the position of the tyrosine. They reported that the medical effectiveness of Gel-gffy adjuvant depends on the amino acid sequence of the peptide [98]. Their immunostimulatory results on bone marrow-derived dendritic cells showed that the Gel-gfyf, Gel-gyff, and Gel-gffy significantly upregulated maturation marker expression. In vivo experiments exhibited that the encapsulation of ovalbumin (OVA) inside the Gel-gfyf, Gel-gyff, or Gel-gffy vaccine triggered robust humoral and cellular immune response, demonstrating the highest immunostimulatory for the Gel-gffy. They showed the importance role of both the sequence and number of aromatic amino acids on the adjuvant potency of Gel-gffy, which provided a great perspective for the structural modifications of peptide hydrogels for the fabrication of new vaccine adjuvants. Su et al. reported the conjugation of antigen epitope with self-assembling peptides to fabricate a nanofibrious hydrogel vaccine platform to target CD8 or CD4 T-cell receptors [99]. The controlled loading of antigens in the nanofibers hydrogel significantly increased the uptake of the antigens, resulting in the activation of dendritic cells (DCs). The immunization in vivo tests in mice indicated that this hydrogel platform did not cause inflammation responses and results in the activation of the MyD88-dependent NF-*κ*B signaling pathway in DCs, which significantly promotes the T-cell immunity. The mice vaccination with this hydrogel induced an enhanced generation of both CD8 and CD4 T cells against EG.7-OVA tumors, without necessity to use other immunoadjuvants. This approach shows the promising potential ability of such hybrid hydrogels for enhancing the efficacy of cancer immunotherapy, compared with free peptide vaccine or aluminum-adjuvant peptide formulation. Anand et al. developed the production of a tag-free recombinant PfMSPFu24 (rFu24) protein expressed in *Escherichia coli* bacteria [100]. This protein is highly immunogenic with the human-compatible adjuvant; thus, they further encapsulated the rFu24 protein inside the two dipeptides hydrogels, based on phenylalanine-*α*, *β*-dehydrophenylalanine (FΔF), and Leucine-*α*, *β*-dehydrophenylalanine (LΔF) to form the effective delivery system. The self-assembly of the peptides was performed spontaneously under physiological conditions to make a stable F∆F and L∆F hydrogels having a high amount of antigen without any change of the three-dimensional (3D) mesh-like structures of the hydrogels after the entrapment of the antigen. The in vivo test in mice showed that these two hydrogels significantly triggered rFu24-specific antibody titers, which demonstrated the great potential of these easily synthesized dipeptide hydrogels for effective antigen delivery. Liu et al. reported that the inoculation of Nap-GdFdFdY (a tetrapeptide hydrogel exhibiting adjuvant effect) can completely protect a nonobese diabetic (NOD) mice from type 1 diabetes (T1D) progress up to 36 weeks age, with better pancreatic islet maintenance with minimal immune cell infiltration [101]. In fact, the peptide induces an enhanced peripheral T regulatory cell (Treg), implicating a “buffering” role of Nap-GdFdFdY on systemic inflammatory tone in NOD mice. These results showed the versatility, applicability, and excellent potency of Nap-GdFdFdY hydrogel as a novel therapeutic intervention for T1D. Delitto et al. applied a neoantigen-targeted hydrogel vaccine to prevent the recurrence of pancreatic adenocarcinoma after incomplete resection in mice [102]. The neoantigen peptides hydrogels (termed PancVax), acted as a stimulator of interferon genes (STING), activated the production of T cells in the draining lymph node, and increased the neoantigen-specific T cells in the spleen. The vaccinated mice showed no residual tumor at two weeks postoperatively, followed by healed incisions at two months with no recurrence. Rudra et al. reported the chemical functionalization of self-assembling Q11 peptide with ovalbumin (OVA) as a potential chemical adjuvant stimulating antibody response [103]. Hang et al. later synthesized several types of breast cancer vaccines and conjugated them with Q11-based peptide [104]. The Q11 domain self-assembles into fibers under mild conditions acting as both adjuvant and the vaccine carrier. These hydrogel-vaccines systems potently stimulate the immune system for generating antibodies against the human breast tumor cells. Wang et al. employed peptide Nap-GFFpY-OMe hydrogels and physically entrapped ovalbumin (OVA) immune adjuvant inside the hydrogel for improving humoral immune responses [105]. The entrapped OVA significantly enhance the antigen (OVA) uptake, which results in an increase in the antibody production and relevant cytokines. Due to the acceptable biocompatibility and degradability of their hydrogels, they can be developed for the formulation of protein vaccines in the immunotherapy of cancer. This research group also reported a chemical conjugation of MUC1 antigen with peptide Nap-GDFDFDYDK to fabricate anticancer hydrogel vaccines, which can initiate both humoral and cellular immune responses [106]. Regarding the hydrogels that can directly trigger the immune response, there are a few reports; for example, Xing et al. synthesized injectable peptide hydrogels by electrostatic coupling of a cationic polypeptide (poly-l-lysine, PLL) and an anionic amyloid dipeptide Fmoc-FF [91]. The PLL/Fmoc-FF can self-assemble into nanofibrous helical hydrogels, stabilized with disulfide bonds. The structure of this hydrogel is somehow like fimbriae, antigens processing immune regulatory adjuvants properties that can activate T cell response for suppressing the growth of B16 cancer cells, both in vitro and in vivo. The results of such studies can demonstrate the promising applications of the peptide hydrogels as immunotherapeutic agents.

Wu et al. employed L/D tetrapeptide hydrogels as the adjuvant for an H7N9 vaccine and reported that the hydrogels can induce a significant antibody production when combined with the H7N9 vaccine [44]. In fact, these L/D tetrapeptide hydrogels enhanced the therapeutic effect of the H7N9 vaccine and could be promising adjuvants for H7N9 vaccines against highly pathogenic H7N9 virus. Yang et al. reported that hydrogels of nonsteroidal anti-inflammatory drug modified D-tetrapeptides (GDFDFDY) are promising candidates for cancer vaccine adjuvants, especially for Fbp-gel and Car-gel [63]. In another study, they reported a novel D-tetrapeptide hydrogel vaccine adjuvant. Antigens were combined with these injectable hydrogels by a vortex or by gently shaking. The vaccines induced strong CD8+ T-cell responses, inhibiting the growth [45]. Zhao et al. used a tetrapeptide hydrogel for the encapsulation of protein to form a hydrogel–vaccine system [43]. Their in vivo tests showed the enhanced antigen uptake capacity and moderate promotion of dendritic cell (DC) maturation. Regarding the in vivo and ex vivo results, they reported the antigen specific antibody creation and splenocyte proliferation, respectively.

### 2.4. Antibacterial and Wound Healing Applications

Nowadays, bacterial infections are considered a common issue in different aspects of life because traditional antibiotics often show low biocompatibility and have drug-resistant effects, resulting in low antibacterial efficacy. Hence, it is vital to develop new materials to reduce infections and improve the quality of life for millions of people all around the world, which can save medical resources and the social economy [1]. Peptide hydrogel-based antibacterial systems are divided into two main groups, including (1) hydrogels with inherent antibacterial activity, usually fabricated from antimicrobial peptides (AMPs), and (2) hydrogel-based delivery systems for synthetic antibiotics (e.g., ciprofloxacin, gentamicin, vancomycin, cephalosporin) (see Figure 3).

The latter category can enhance the therapeutic efficacy of antibiotics and decrease undesired toxicity [107]. In 2022, Huang et al. reported antibacterial reticulated NZ2114 peptide hydrogels containing hydroxypropyl cellulose (HPC) and sodium alginate (SA) [108]. They studied the sustained release of these hydrogels with 97.88 ± 1.79% and 91.10 ± 10.52% rate (in 72 h), for cellulose- and alginate-based hydrogels, respectively. Both hydrogels showed good short-term cytocompatibility and hemocompatibility; however, the HPC-NZ2114 hydrogel demonstrated higher mechanical strength with a smaller pore size, compared to the SA-NZ2114 hydrogel. Both in vitro and in vivo tests confirmed the efficient antimicrobial activity of both HPC/SA-NZ2114 hydrogels. Skin model experiments showed that the wound closure properties of HPC-NZ2114 hydrogel are higher than those of the SA-NZ2114 hydrogel and antibiotic groups on day 7, demonstrating that the HPC-NZ2114 hydrogel can be an interesting candidate for infection wound dressing applications. Wu et al. developed a pH-sensitive hydrogel containing peptide DP7 (VQWRIRVAVIRK) and oxidized dextran (DP7-ODEX hydrogel) exhibiting dual functionalities: (1) having inherent antibacterial effects and (2) ability to deliver antibiotics [109]. The antimicrobial peptide, DP7, can combine with many antibiotics; thus, a common antibiotic, ceftazidime, was encapsulated into the DP7-ODEX hydrogel and showed a synergistic antimicrobial effect against multi-drug-resistant *P. aeruginosa*. In addition, due to the presence of suitable aldehyde groups in this hydrogel structure (in oxidized dextran), the hydrogel can easily attach to amine groups of the wound tissue and expand on the irregular surface of skin defects, facilitating the adhesion of epithelial cells and acting as an accelerating wound-healing material. Xie et al. developed an inherent antibacterial and injectable hydrogel, based on a heterochiral peptide C16-DL4LR4, demonstrating optimal time-dependent antibacterial activity against both Gram-positive (*Staphylococcus aureus* (*S. aureus*) and *Bacillus subtilis* (*B. subtilis*)) and Gram-negative (*Escherichia coli* (*E. coli*) and *Shigella sonnei* (*Sh. Sonnei*)) bacteria [110]. They reported a significant decrease in the survival rate (after 5 h) of *S. aureus*, *B. subtilis*, *E. coli,* and *Sh. sonnei* bacteria to 5.9%, 0%, 5.3%, and 11.09%, respectively. The chiral nanofibers (15–20 nm) of the hydrogel were responsible for the antibacterial effect by the aggregation of bacteria and destruction of the cell membrane. Additionally, the in vitro experiments revealed the high biocompatibility of this hydrogel on the mouse embryonic fibroblasts. The results of this research are useful for the development of the inherent antibacterial hydrogels based on the heterochiral peptides. Xiong et al. reported a hyaluronic acid (HA)-hydrogel containing Fe^3+^ ions and a dopamine-modified antimicrobial peptide (DAP), possessing photothermal-assisted wound-healing characteristics for bacterial infection [111]. The inherent antimicrobial DAP peptide acts as an ROS-scavenger and can reinforce the hydrogel structure by forming Schiff’s base and coordination bonds with aldehyde hyaluronic acid (AHA) and Fe^3+^, respectively. The in vitro and in vivo results supported that this hydrogel possesses an accelerated wound-healing property on the infected site, demonstrating the remarkable potential antibacterial ability of hydrogels having antioxidant capability and photothermal effects. Xu et al. developed a new Fmoc-FFWDD-OH peptide hydrogel formed by rare earth metal ions, (Tb^3+^, Eu^3+^, and La^3+^) [112]. This hydrogel demonstrated excellent biocompatibility and showed elasticity-dominating properties, as well as a good antibacterial performance on *Escherichia coli* (*E. coli*). Yuan et al. reported a one-pot synthesis of a cationic peptide hydrogel, which is salt-responsive with antifouling and bactericidal character, and contains *ε*-poly-L-lysine, poly (ethylene glycol) diglycidyl ether, and poly(DVBAPS-co-GMA) via a one-pot method [113]. The structure of these biocompatible hydrogels has the flexibility to quaternize for enhancing the antibacterial property. These tripeptide hydrogels demonstrated high antibacterial properties (>90%), cell adhesiveness, and proliferation efficacy (~400%) with a very high sterilization rate (up to ~96%). Liu et al. developed an antifouling hydrogel coating via the co-polymerization of acrylic acid and sulfobetaine methacrylate on a polymeric substrate followed by the embedding of antimicrobial peptides (AMPs) with the sequences WRWRWR-NH_2_ or RLARIVVIRVAR-NH_2_ [114]. This hydrogel coating revealed a good hemocompatibility with excellent bactericidal and adhesion properties on both Gram-negative and Gram-positive bacteria. It should be mentioned that the hydrogel coating could guarantee the long-term bioactivity of AMPs, and it well retains the antifouling hemocompatibility capacity of the bulk hydrogels. The animal microbiological experiments verified the anti-infection performance and an excellent anti-inflammatory function of this surface, providing a perspective view for the prevention of bacterial infections of blood-contacting devices and related instruments. Wei et al. employed an antibacterial hydrogel system for anti-infection of chronic wounds, containing the antimicrobial peptides (inflammatory modulator) and platelet-rich plasma (PRP) (to enhance collagen deposition) [115]. The ODEX/HA-AMP/PRP hydrogel was fabricated under physiological conditions by combining oxidized dextran (ODEX), antimicrobial peptide-modified hyaluronic acid (HA-AMP), and PRP. The hydrogel exhibited antibacterial effects against three pathogenic bacterial strains (*E. coli*, *S. aureus* and *P. aeruginosa*) and could enhance the proliferation and migration of L929 fibroblast cells. The in vivo experiments further demonstrated a significant wound-healing property of this hydrogel in a diabetic mouse infection through regulating inflammation and enhanced collagen deposition and angiogenesis. Trusek et al. used a peptide linker (Leu-Leu-Gly) to attach graphene oxide (GO) to the model drug, amoxicillin (AMOX) [116]. To fabricate this system, the authors dispersed GO, AMOX, and a bromelain plant enzyme (responsible for releasing AMOX from GO) into the hydrogel. The drug release could happen by the enzymatic hydrolysis of the peptide bond in the linker. The antibacterial activity of the hydrogel was tested on the *Enterococcus faecalis* strain, responsible for periodontal and root canal diseases. The in vivo biocompatibility of this carrier was confirmed on the mouse fibrosarcoma cell line WEHI 164. They proposed that this hydrogel can be considered as a suitable candidate for the treatment of inflammation in dentistry. Afami et al. studied antimicrobial properties of an ultrashort peptide hydrogel, (naphthalene-2-ly)-acetyl-diphenylalanine-dilysine-OH (NapFFεKεK-OH), in both solution and hydrogel form against *Staphylococcus aureus*, *Enterococcus faecalis,* and *Fusobacterium nucleatum* and also showed the biocompatibility of this hydrogel on dental pulp stem/stromal cells (DPSCs) [117]. They reported that the NapFFεKεK-OH hydrogels have antimicrobial activity against *E. faecalis* and *F. nucleatum*, while the peptides in solution were effective against biofilms of *S. aureus* and *E. faecalis*. Marchesa et al. reported non-covalent loading of the hydrophilic antibiotic ciprofloxacin inside a self-assembling tripeptide D-Leu-Phe-Phe forming together supramolecular nanostructures [118]. Interestingly, the hydrogel exhibited moderate antibacterial activity against Gram-negative bacteria, and it is highly compatible with human red blood cells, as well as mouse L929 fibroblast cell cultures. Due to high drug loading efficiency and controlled drug release property, this hydrogel-ciprofloxacin system triggers significant antimicrobial activity against *Staphylococcus aureus* and *Escherichia coli*.

Regarding the inherent antibacterial hydrogels, they are well-known as promising candidates for clinical use against antibiotic-resistant bacterial infections. They usually have an amphiphilic character with a hydrophobic core and positively charged surfaces, facilitating electrostatic interactions between the hydrogels and negatively-charged membranes of bacteria. These hydrogels exhibit several antibacterial mechanisms including the disruption of bacterial membranes to inhibit or alter the biochemical reactions of the bacterial cell [119]. Salick et al. synthesized a MAX1 peptide *β*-hairpin hydrogel exhibiting inherent antibacterial activity on the outer surface of the hydrogel against Gram-positive (*Staphylococcus epidermidis*, *Staphylococcus aureus*, and *Streptococcus pyogenes*) and Gram-negative (*Klebsiella pneumoniae* and *Escherichia coli*) bacteria, all prevalent in the clinical setting [120]. More importantly, the hydrogel showed the hemocompatibility on human erythrocytes. Due to the high arginine residue content of this hydrogel, it interacts with phosphorus atoms of the bacterial membrane through hydrogen bonds. Schmidt et al. reported an antibacterial hydrogel in which they prepared arginine-rich hydrogels (VKVRVRVRVDPPTRVRVRVKV), demonstrating a high effectiveness against both Gram-negative and Gram-positive bacteria, especially multi-drug resistant *Pseudomonas aeruginosa* [121]. Nandi et al. reported a series of peptide hydrogels processing different chain lengths of alkyl groups [122]. Although they did not employ lysine or arginine residues in the hydrogel structure, their results showed high antimicrobial property against *Staphylococcus aureus*, *Bacillus subtilis*, and *Escherichia coli*. These hydrogels demonstrated no cytotoxicity for human lung carcinoma and showed a little incompatibility for human red blood cells at the same concentration used for the antibacterial effect. The antibacterial character of these hydrogels is due to the presence of hydrophobic moieties of the hydrogel, such as to the structure of natural antibacterial peptides.

Malhotra et al. reported the broad-spectrum antibacterial activity of dipeptides Boc-D-Phe-*γ*4-L-Phe-PEA and Boc-L-Phe-*γ*4-L-Phe-PEA against several bacteria, such as *Escherichia coli*, *Pseudomonas aeruginosa*, *Bacillus subtilis*, and *Staphylococcus aureus* [123]. Antibacterial properties of these two hydrogels were related to the entrapment of bacteria in the gel networks and interaction with cell membrane components and lysis. Their results showed that both gels are compatible with NIH 3T3 mouse embryonic fibroblast cells (mammalian), indicating the potential ability of the *α*/*γ* hybrid peptides in preventing the infections. Wan et al. synthesized a class of lysine residue-modified cationic supramolecular hydrogels showing both pH-responsive and antibacterial properties [124]. As mentioned above, there are hydrogels exhibiting inherent antibacterial activity, fabricated from the antimicrobial peptides (AMPs). However, most AMPs show low water solubility and are unable to form well-defined hydrogels. To overcome this issue, hydrogelation can be performed under enzymatic catalysis to enhance the cross-linking of small peptides. Bai et al. employed lysyl oxidase (LO) and plasma amine oxidase (PAO) enzymes for the self-assembly of the synthetic antibacterial peptide A9K2 into the fibrous antibacterial hydrogels [125]. The antibacterial property of this A9K2 hydrogel was studied on culturing bacteria (*B. subtilis* 168), and high antibacterial activity was reported. Additionally, the cytotoxicity of this hydrogel was studied on the mammalian cells, NIH 3T3 and HEK293, and low cell death was detected, suggesting low toxicity of this A9K2 hydrogel. More importantly, cell selectivity of the hydrogel was tested by the coculturing of *B. subtilis* 168 with NIH 3T3 cells, with and without this hydrogel. In the absence of hydrogel, the bacteria caused a significant death of the mammalian cells; conversely, the hydrogel can prevent the bacterial contamination of the NIH 3T3 cells (spread out on the hydrogel surface, with no morphological change), indicating the high cellular selectivity of the A9K2 hydrogel. Bai et al. employed the enzymatic catalytic strategy to synthesize an amphiphilic I_4_K_2_ peptide hydrogel (I: isoleucine), demonstrating a significant antibacterial property against both Gram-positive and Gram-negative bacteria, with low cell toxicity [126].

Due to the antibacterial and biocompatible properties of the peptide hydrogels, they have been studied as wound-dressing materials due to their ability to provide a wet permeable environment to delivery drugs and gases, resulting in rapid wound closure and satisfactory recovery. Carrejo et al. synthesized a K_2_(SL)_6_K_2_-multidomain peptide to accelerate the wounds-healing process of a diabetic mice model through increasing granulation tissue formation, re-epithelialization, vascularization, and innervation [127]. Their results showed the potential application of such peptides for the treatment of wounds in diabetic patients. Loo et al. used ultrashort aliphatic peptides, containing 3–7 natural aliphatic amino acids, as exemplified by Ac-LIVAGK-NH_2_ [128]. Due to their amphiphilic character, these peptides self-assemble into macromolecular nanofibrous hydrogels (cross-linked hydrogel) which showed faster healing of burn wounds, compared to the clinically used Mepitel^®^ (Molnlycke Health Care Srl, Gallarate (VA), Italy)), a silicone-coated polyamide. In detail, they employed two peptide concentrations of 5 and 7.5 mg/mL achieving 86.2% and 92.9% wound closure, respectively (much more effective than the Mepitel^@^ with 62.8%). In another study, they synthesized disulfide cross-linking hydrogels using cysteine-containing ultrashort peptides that exhibited a higher wound healing ability than the non-cross-linked hydrogel [129]. Yergoz et al. used the biodegradable heparinmimetic hydrogel for burnt wound healing [130]. The bioactive hydrogel scaffolds accelerated angiogenesis due to the affinity of this type of hydrogel for basic growth factors. Compared with the commercial wound dressing 3Mt Tegadermt and the non-bioactive peptide hydrogel, this heparin-mimetic hydrogel promoted the wound healing process through the formation of well-defined and the collagen-rich granulation tissue layers with a high density of new blood vessels. This resulted in an increased re-epithelialization and skin repairing progress with minimal crust formation. Their results pave the way towards the development of bioactive hydrogel as a wound-dressing scaffold to accelerate wound healing. Ren et al. synthesized several Fmoc-functionalized dipeptides hydrogels and applied them in the self-healing applications as an endoscopic submucosal dissection (ESD) filler to facilitate the operation compared with clinical ESD fillers normal saline, sodium hyaluronate, and a commercial polymer PF-127 [22]. They studied the synergetic effect of hydrophobic and hydrogen-bonding interactions on the mechanical strength and self-healing properties of hydrogels. This enhancement of hydrophobic interactions significantly improved the mechanical rigidity and hydrogen bonds, which facilitated the self-healing property. The in vivo tests in mice and living mini-pigs proved that this injectable Fmoc-YL hydrogel, with shear-thinning and instantaneous self-healing characteristics, could be employed as an ideal ESD filler with a longer workability, higher stiffness, and lower inflammatory response, facilitating the operation and avoids leakage, compared with the mentioned clinical ESD fillers.

Banerjee et al. synthesized a modified dipeptide hydrogel showing antibacterial activity against Gram-negative bacteria *Escherichia coli* and *Pseudomonas aeruginosa* [26]. They reported the enzymatic stability (inside the cells) and negligible toxicity (on human erythrocytes and human fibroblast cells) of this hydrogel. Das et al. introduced a class of biocompatible dipeptide hydrogel showing antibacterial activity against Gram-positive and Gram-negative bacteria [23,24]. By varying the aromatic groups in the structure of dipeptides, the antibacterial activity and cell toxicity of the hydrogels changed.

### 2.5. 3D Bioprinting and Tissue Engineering

Tissue engineering and 3D bioprinting are interconnected multidisciplinary fields dealing with the fabrication of biocompatible materials for the repairs or replacement of unnormal tissues/organs. These scaffolds, used for tissue engineering, should be highly compatible with the cells inside the body. Peptide hydrogels can protect cells from deactivation and facilitate their nutrient transport, providing excellent biological environment for cell adhesion [131]. They have shown interesting platforms for the culture of a number of cell lines promoting biological processes for the cell growth. Due to a high biocompatibility of the peptide hydrogels, they can mimic the natural extracellular matrix (ECM) for endogenous cell growth. More importantly, the degradation of these hydrogels results in amino acids that are safe and will not induce side effects or immune or inflammatory reactions inside the body [1]. In 2022, Li et al. developed a novel self-healing and injectable hydrogel (GMO hydrogel) containing loaded KP and QK peptides to improve the vascularized regeneration of minor, irregular defects of bone [132]. In this hydrogel, the imine bonds between gelatin methacryloyl and oxidized dextran provides an excellent injectability and fittability. They entrapped GelMA inside the hydrogel followed by the photopolymerization of GelMA to enhance the mechanical strength of this scaffold, suitable for bone regeneration. The in vitro results showed a dramatic enhancement of the osteogenic differentiation of bone marrow mesenchymal stem cells (BMSCs) and angiogenesis ability of human umbilical vein endothelial cells (HUVECs). Moreover, they studied the bone formation in rat calvaria and reported a high synergistic effect of the loaded KP and QK peptides to promote the new bone formation, which demonstrates the potential ability of this composite hydrogel as an efficient safe scaffold for the vascularized bone regeneration. Ye et al. conjugated Ac-(RADA)_4_-CONH_2_ (RAD) peptide with TGF-*β*1-simulating peptide LIANAK (CM) to obtain the modified self-assembling peptide Ac-(RADA)_4_-GG-LIANAK-CONH_2_ (RAD-CM) [133]. The resulting hydrogel promoted expressions of chondrogenic genes and extracellular matrix deposition, which then was combined with decellularized cartilage extracellular matrix (DCM), forming a scaffold for articular cartilage repair. This composite scaffold possesses good bioactivity and structural features to improve the neocartilage restoration and the reconstruction of the osteochondral unit, which provides a promising opportunity for in situ cartilage regeneration due to the presence of TGF-*β*1-simulating peptide in the structure of the hydrogel. Zheng et al. developed a dual crosslinking conductive hydrogel containing modified hyaluronic acid (HA), gelatin (G), and Fe^3+^, demonstrating self-healing, degradable, and acceptable mechanical features adapted to the myocardial systolic-diastolic cycle resulting in a significant enhancement of the myocardial wall thickness, very similar to that of normal myocardium (by one time injection) [134]. This hydrogel contains a chemically conjugated-peptide sequence of GGR-KLT and was used for the encapsulation of bone mesenchymal stem cells (BMSCs), injected in a rat MI model. The in vivo results showed three benefits of this system including: (1) efficient prevention of excessive matrix degradation, (2) induced release of KLT for at least 14 days, resulting in a dramatic improvement of angiogenesis, and (3) enhancement of myocardial regeneration during a 28-day period. Their results confirmed the comprehensive recovery and regeneration of the injured myocardium, demonstrating a great perspective of this hydrogel for clinical applications. Halperin-Sternfeld et al. conjugated the sulfated polysaccharide with the fluorenylmethoxycarbonyl diphenylalanine (FmocFF) peptide hydrogelator to enhance the mechanical properties of the polysaccharide [135]. This injectable hydrogel showed flexible mechanical properties with a nanofibrous structure, mimicking the extracellular matrix environment. The composite hydrogels were used for the controlled release of curcumin, and they supported MC3T3-E1 preosteoblasts viability and calcium deposition, bearing a promising candidate as new tunable scaffolds for tissue engineering and cell differentiation applications. Zhan et al. linked multi-component co-assembly peptides with conductive polypyrrole (PPy) to obtain an injectable ROS-scavenging/conductive composite hydrogel (R&C-Gel) [136]. The strong ROS-scavenging ability of this hydrogel was achieved by integrating antioxidant TEMPOL into the peptide. This antioxidant and conductive hydrogel demonstrated efficient ROS removal in cardiomyocytes and decrease apoptosis under oxygen-glucose deprivation (OGD) conditions. From the evidence of expression of Cx-43 protein and the rhythmic intracellular Ca^2+^ puffs, the authors concluded that this hydrogel could enhance the contractile and electrical properties of cardiomyocytes. The results revealed that the combination of R&C-Gel with cardiomyocytes can largely improve the cardiac repair and rebuilt cardiac function, and thus this research gained insight into the synthesis of multifunctional conductive hydrogels specified for the cardiac tissue engineering. Zhang et al. fabricated a RATEA16 hydrogel scaffold for the drug loading of rhVEGF165 and BMP-2 for the in vitro vasculogenic/osteogenic differentiation of human umbilical vein endothelial cells (HUVECs) and human stem cells of the apical papilla (SCAPs) [137]. They introduced this system as an injectable self-assembled RATEA16 hydrogel for efficient drug delivery showing a sustained release of VEGF and BMP-2 to facilitate the vascular and bone formation. Han et al. synthesized a new functionalized peptide RAD/SA1 by the conjugation of RADA16-I with Sa12b (as an inhibitor of acid-sensing ion channels) [138]. Their structural results confirmed the self-assembly of RAD/SA1 peptide into 3D nanofiber hydrogel scaffolds at acidic pHs. The human nucleus pulposus mesenchymal stem cells (hNPMSCs) cultured in the 3D scaffolds revealed an extremely low cytotoxic effect of both RADA16-I and RAD/SA1. Additionally, the RAD/SA1 improved the proliferation of hNPMSCs, compared to that in culture plates and pure RADA16-I. Additionally, this RAD/SA1 peptide inhibited the p-ERK expression through Ca^2+^-dependent p-ERK signaling pathways, showing that this peptide hydrogel with the short motif of Sa12b is a suitable scaffold for nucleus pulposus tissue engineering. Chai et al. used a peptide hydrogel scaffold to study its effect on the improvement of angiogenesis, inhibition of the differentiation and adhesion of keratinocytes, as well as reduction of the formation of glial scar tissue [139]. This scaffold showed the promotion of angiogenic factors expression and decreased the secretion of pro-inflammatory factors to a certain extent, as well as showed an inhibition of glial scar tissue formation and could repair damaged tissue, suitable for tissue engineering of the central nervous system (CNS). Zhang et al. employed a dual-functionalized peptide hydrogel containing the brain-derived neurotrophic factor and vascular endothelial growth factor mimetic peptides RGIDKRHWNSQ (RGI) and KLTWQELYQLKYKGI (KLT) for the fabrication of an artificial neurovascular microenvironment [140]. The in vitro results showed that this hydrogel can promote the outgrowth of pheochromocytoma (PC12) cells and improve the formation of tube-like structures of human umbilical vein endothelial cells (HUVECs). The in vivo tests in a rat brain injury model showed the rapid promotion lesion infiltration of neural and vascular cells, and this hydrogel can mediate neurovascular crosstalk by regulating secretion of paracrine factors from PC12 cells and HUVECs. By co-culturing both of these cells on this hydrogel, cell–cell communication was enhanced significantly, followed by the acceleration of PC12 differentiation with an increase in pseudopodia and formation of HUVECs’ tube-like structure. Hivare et al. used a DNA based hydrogel that was coated on a glass and chemically conjugated with a synthetic IKVAV peptide [141]. They reported that this hydrogel triggered the enhanced differentiation of the neuronal movement of cytoskeleton and microtubules, and it prolonged neurite length, and modified endocytic mechanisms in neuroblastoma-stem cells, compared to the unmodified DNA hydrogel and controls. They proposed this hydrogel as a promising composite scaffold for development of nerve tissue conduits in neuroscience and neuro-regeneration fields. Sharma et al. explored the noncovalent interactions between the sugar and peptide-based building blocks to develop advanced tissue scaffolds [142]. They employed a short ionic complementary peptide, Nap-FEFK, and TEMPO-oxidized nanofibrillar cellulose (TO-NFC) to construct tunable hydrogels. Interestingly, with the change of TO-NFC/Nap-FEFK ratio, the cellular response of this composite scaffolds was modified, with an optimum 10:1 (*w*/*w*) ratio of TO-NFC: Nap-FEFK for the maximum cellular survival and proliferation, which was similar to that of Matrigel in three-dimensional culture conditions, demonstrating a great potential of this hydrogel to be used in advanced tissue engineering applications. Moreover, the hydrogel showed no inflammatory response in Raw cells and supported their survival and proliferation. Rosa et al. demonstrated the combination of Lys residue with an Fmoc group or with the fluorenyl methoxycarbonyl-diphenylalanine (FmocFF) to obtain seven novel hydrogels [143]. Their in vitro studies demonstrated a high biocompatibility of hydrogels (>95%) over 72 h of cell treatment, and they reported the favoring adhesion, spreading, and growing of 3T3-L1 cells cultured on one of the tested hydrogels. Liu et al. combined RAD and Dentonin to fabricate a scaffold, based on the self-assembled peptide RAD/Dentonin hydrogel, to promote pulp regeneration [144]. The results showed a *β*-sheet-based nanofiber network structure for RAD/Dentonin hydrogel, used as a new scaffold with superior biocompatibility and enhanced adhesive proliferation, odontogenic differentiation, and mineralization deposition of human dental pulp stem cells (hDPSCs). They introduced this hydrogel scaffold as a promising candidate for endodontic tissue engineering. Thomas et al. developed a hydrogel possessing a dynamic interpenetrating network suitable for supporting chondrocyte growth and differentiation [145]. For this purpose, they used an amyloid-inspired peptide amphiphile self-assembled into nanofibers and embedded in a chemically cross-linked polysaccharide network of carboxymethyl cellulose dialdehyde (CMC-D) and carboxymethyl chitosan (CMCh). The adaptable structure of the hydrogel facilitated various cellular functions while keeping its integrity. The hydrogels were used to support in the vitro chondrogenesis of human chondrocytes, and the authors reported the improved cell growth and cartilage-specific ECM production useful for cartilage tissue engineering. Yaguchi et al. employed a cell-adhesive, fiber-forming peptide hydrogel to imitate the jigsaw-shaped hydrophobic surface in the dovetail-packing motif of glycophorin A as an artificial extracellular matrix for regenerative therapy [146]. This hydrogel provided controlled release of vascular endothelial growth factor, and their results on a subacute-chronic phase mouse stroke model revealed the cell transplantation-free regenerative therapeutic effects, highlighting a great potential of this hydrogel for using in injured tissue regeneration. Alheib et al. used gellan gum (GG)-based hydrogels tailored with a laminin-derived peptide for the differentiation of myoblasts and developing a skeletal muscle platform [147]. For this reason, they attached murine skeletal muscle cells (C2C12) to several different laminin-derived peptides (CIKVAVS (V), KNRLTIELEVRTC (T), and RKRLQVQLSIRTC (Q)). Additionally, they studied the binding of chemically functionalized GG to the laminin-derived peptides. All the hydrogels demonstrated the cellular adhesion; however, only peptide Q-functionalized hydrogels showed spreading and differentiation. Dufour et al. combined a new self-assembling peptide (referred as IEIK13) with articular chondrocytes treated with a chondrogenic cocktail (BMP-2, insulin, and T3, designated BIT) to study the repair of full-thickness cartilage defects in cynomolgus monkey [148]. Their in vitro experiments showed that IEIK13 hydrogel could facilitate the cartilage production by monkey articular chondrocytes treated with BIT. These IEIK13 implants (loaded with or devoid of chondrocytes) showed the restoration of full-thickness cartilage injuries, similar to cartilage-characteristic regeneration. Wiseman et al. studied the ability of Fmoc-DIKVAV as scaffold and vehicle for grafted cells (in adult Fischer 344 (F344) rats) after mild thoracic contusion spinal cord injury (SCI) (thoracic level 10 [T10]) [149]. Their results showed that, after contusion SCI, Fmoc-DIKVAV hydrogel scaffold facilitated the cellular infiltration and axonal regrowth. Zanotto et al. combined a self-assembling KLD peptide hydrogel (functionalized with PDGF-BB and HB-IGF-1evaluate growth factors) with trypsin pretreatment to study the microfracture healing of a critically sized cartilage defect in horses subjected to strenuous exercise [150]. Compared with microfracture alone, they reported the improvement of cartilage healing and better integration of the newly synthetized cartilage. Regarding the hydrogel-based bioinks, Jang et al. conducted a great work in which gelatin methacrylate (GelMA) hydrogel was incorporated with a vascular endothelial growth factor (VEGF) peptide, and then it was printed using a 3D bio-printer [151]. The VEGF peptide showed biocompatibility on mouse fibroblasts cell lines (NIH 3T3) and human umbilical vein endothelial cells, promoting cell viability, proliferation, and tubular structure formation. In addition, successful results were obtained on the wound-healing efficacy of this hydrogel on a pig skin wound model. Zhang et al. prepared integral active oyster peptide microspheres (OPM) (as the filler) and combined them with catechol functionalized chitosan (CS–C) (as the polymeric matrix) and *β*-sodium glycerophosphate (*β*-GP) (as a thermal sensitizer) to fabricate a thermosensitive hydrogel CS-C/OPM/*β*-GP [152]. The authors tested this hydrogel for wound healing applications, and the results indicated that the hydrogel could accelerate the migration of L929 cells. The experiments on the mouse skin wound demonstrated the ability of this hydrogel to inhibit the aggregation of diversified inflammatory cells and facilitate the collagen fibers generation and new blood vessels of the wound, as well as the biosynthesis of total protein (TP) in granulation tissue and regulating the expression of Ki-67 and VEGF in the injury, resulting in the rapid wound healing. They also reported the negligible toxicity of the hydrogel on L929 cells, introducing this hydrogel as a promising medical dressing for wound healing. Roy et al. used tHA-PEGDA hydrogel for the incorporation of RGD adhesion peptides and an anti-vascular endothelial growth factor receptor-2 (VEGF-R2) DNA to obtain a bifunctional hyaluronic acid hydrogel [153]. RGD peptides improved growth of cells, while the anti-VEGF-R2 DNA aptamer seemed to promote cell viability, and also angiogenesis by tube formation by endothelial cells, useful for advanced wound healing applications. Okuno et al. tested the potential applications of a hydrogel scaffold KI24RGDS to facilitate meniscal regeneration in an induced rabbit meniscal defect model [154]. They reported a high biocompatibility of the KI24RGDS hydrogel with strong stiffness, possessing a similar native extracellular matrix useful for cell adhesion and proliferation. Their results demonstrated that this hydrogel could promote the repair and regeneration in a rabbit meniscal defect model. For the bone tissue formation, Panek et al. loaded dexamethasone (DEX) in a hydrogel-based scaffold to prepare (DEX)-loaded [COCH_3_]-RADARADARADARADA-[CONH_2_] (RADA 16-I) scaffold showing the optimal release of DEX concentration under perfusion force [155]. The best-engineered bone tissue is regenerated on this hydrogel scaffold with 4 × 10^−^^4^ M of loaded DEX with a perfusion rate of 0.1 mL/min for 21 days, demonstrating a high potential for application in regenerative orthopedics. Koutsopoulos et al. employed peptide hydrogels for the encapsulation of neural stem cells [156]. These encapsulated cells could proliferate and differentiate for 14 days and thrive for at least five months. More importantly, the 3D matrix of these hydrogels provided suitable space and nutrients for the cell growth, indicating the potential ability of the hydrogel for practical applications in tissue engineering. Tang et al. synthesized pH-sensitive pentapeptides hydrogel in which, by variations in solvent accessibility, the *β*-sheets are increased resulting in the rapid gelation of these peptides, which protects the inner cells from acute membrane disruption [50]. These injectables hydrogels provided the cytocompatible encapsulation of oligodendrocyte progenitor cells in vitro and can be injected, inducing the proliferation of cells in vivo. Zamuner et al. proposed the chemoselective ligation between a self-assembling peptide (called EAK) and three different bioactive molecules including: (1) an adhesive sequence with 4 Glycine-Arginine-Glycine-Aspartic Acid-Serine-Proline (GRGDSP) motifs per chain, (2) an adhesive peptide mapped on h-Vitronectin, (3) and the growth factor Insulin-like Growth Factor-1 (IGF-1) [157]. These three hydrogels showed a significant increase in the adhesion and proliferation of mesenchymal stem cells and supporting cell spreading and elongation, demonstrating these three scaffolds as a very promising 3D model for cell cultures. Stüdle et al. used TGF-*β*_3_ or BMP-2 functionalized PEI-peptide hydrogels for the encapsulation of human bone marrow-derived mesenchymal stromal cells (BMSCs) and nasal chondrocytes (NCs) by enzymatical polymerization in vivo [158]. In these hydrogel scaffolds, BMSCs could form ossicles consisting of bone and marrow by endochondral ossification, while NCs formed cartilage tissue to maintain phenotypic stability. Regarding the 3D printing, peptide hydrogels are of great significance as bioinks; for instance, Paphael et al. synthesized a peptide hydrogel and used it as a bioink for 3D cell bioprinting [159]. They utilized a commercial extrusion-based AM system for the fabrication of different biological scaffolds by direct-printing EpH4 cells containing peptide gelators. These encapsulated cells could grow and proliferate after 7 days of culture. Echalier et al. fabricated a bio-printable peptide hydrogels by the combination of silylated PEG and GRGDSP) peptide (P represents proline) dissolved in DPBS buffer (at an appropriate ratio), which results in the formation of siloxane bonds [160]. This printed scaffold demonstrated a high biocompatibility suitable for culturing the mesenchymal stem cells. Loo et al. used an ultrashort peptide hydrogel for a printable scaffold for 3D cell culture [161]. They used self-assembling lysine-containing hexapeptides to prepare extremely rigid hydrogels (40 kPa) exhibiting suitable shape fidelity in PBS^−^ buffer to form hydrogel droplet arrays. They injected Human H1 embryonic stem cells into these hydrogel droplets and reported the successful cell proliferation into 3D spheroids. These cells expressed various pluripotent nuclear transcription factors and surface biomarkers, indicating the potential ability of this hydrogel in tissue engineering and 3D bioprinting. Additionally, onto specific domains of these hydrogel droplet array, they reported multi-cell culture by sequentially depositing human umbilical vein endothelial cells, fibroblasts, and keratinocytes. After 7 days of the co-culturing, multicellular and multidomain constructs were obtained that potentially could be used as an in vitro skin model. In addition, they reported the long-term cell culture for organotypic constructs with high in vivo biocompatibility and stability. For instance, Caco2 cells injected into the hydrogel survived up to 20 days keeping their normal functions. Regarding the in vivo tests, it is worth mentioning that the healthy C57BL/6 mice persisted for at least 2 months with no immune response. Koutsopoulos et al. cultured the neural stem cells inside the hydrogel scaffolds in the serum-free condition and reported the 3D differentiation of these neural tissues into progenitor neural cells, neurons, astrocytes, and oligodendrocytes, which could thrive for at least five months [156]. Sheikholeslam et al. employed a hybrid hydrogel containing peptide (EFK8)-carbon nanotube (SWNT) for tissue and 3D tumor engineering [162]. The combination of SWNTs with the hydrogel enhanced the cell adhesion, proliferation, and growth of NIH-3T3. Additionally, they reported the promotion of morphological change of the cultured A549 lung cancer cells from spheroidal to stretched. Sleep et al. utilized an amphiphilic peptide hydrogel as scaffolds for the transplantation of muscle stem cell [163]. The stiffness of these scaffolds was suitable for the determination of the macroscopic degree of cell alignment. These scaffolds provided a compatible biological media for the myogenic progenitor cell survival and maturation. In another study, they used this peptide containing laminin mimetic IKVAV (V: valine) sequence to form bioactive supramolecular nanofibers for neural transdifferentiation [164]. These nanofibers could initiate the formation of neuroectodermal lineage commitment after 1 week, followed by promoting the expression of several main biological components including: neuronal markers b-III tubulin (TUJ-1), microtubule-associated protein-2 (MAP-2), neuronal nuclei (NEUN), and the extracellular matrix laminin (LMN) after 2 weeks. Cheng et al. reported a silk fibroin (SF)-based hydrogel containing NapFF peptides for the tissue engineering applications [165]. The self-assembling NapFF peptides could facilitate the formation of *β*-sheet structure of the silk fibroin via hydrophobic and hydrogen bonding interactions. To enhance cell adhesive behavior of this scaffold, they modified NapFF with a GRD moiety (NapFFRGD) to form a peptide-SF-based hydrogel (Gel RGD). The cell attachment and spreading properties were improved by seeding HUVECs on the surface of Gel RGD, resulting in the good cytocompatibility of the hydrogel. Due to the significance of vascularization in tissue regeneration, they further studied the potential application of the Gel RGD matrix for promoting the microvasculature formation. For this purpose, they used VEGF, as a major regulator of vascularization, placed with HUVECs on the surface of Gel RGD. This resulted in the formation of an extensive tubular network within 72 h. For the in vivo evaluation, the VEGF-loaded hydrogel implanted into the dorsal side of mice to check the microvascularization, with and without the hydrogel. Compared to the control test without the hydrogel, the VEGF-containing hydrogels promoted the formation of more blood capillaries near the implanted zone composed of endothelial cells. Alshehri et al. reported the use of two different tetrapeptide hydrogels as scaffold for the growth and osteogenic differentiation of human mesenchymal stem cells. Their results showed that the presence of positively charged Lysine amino acid in these hydrogels positively influence on the cell adhesion and spreading [41].

## 3. Conclusions

This review showed that peptide-based hydrogels have been extensively studied [13,166] and recently developed for biosensing, anticancer drug delivery, vaccines, antimicrobial and wound healing materials, 3D bioprinting, and tissue engineering. The literature exhibits them as promising candidates for improving traditional methods in diverse biomedical applications. However, for their clinical trials, more studies are needed to fully fit all the requirements of a clinical material in terms of physicochemical and biological properties. So, this is still a frontier research area that is supported by the worthwhile knowledge provided by previous publications. Future studies can be directed to the development of green and cost-effective industrial peptide hydrogels by considering the enhancement of workability in the biological field. For this goal, it is necessary to improve the intelligence of peptide-based hydrogels by generating: (1) smart hydrogels which are environmentally friendly and sensitive to different stimuli to improve the biological value of hydrogels; (2) multifunctional peptide-based hydrogels based on theoretical knowledge of structural biology and bioinformatics, and (3) bioactive hydrogels prepared by genetic engineering with the key direction of future development. On this basis, we believe that peptide-based hydrogels will be an important research topic with great vitality and practical potential in biomedical applications.

## Figures and Tables

**Figure 1 materials-15-05871-f001:**
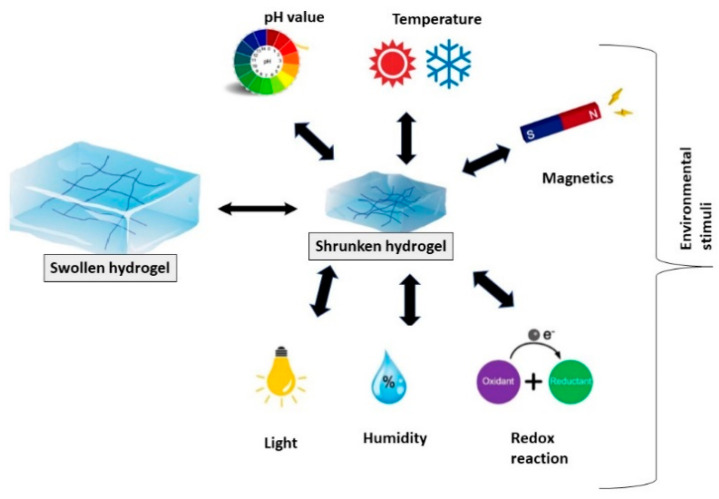
Schematic illustration of hydrogel biosensing application as a function of external stimuli.

**Figure 2 materials-15-05871-f002:**
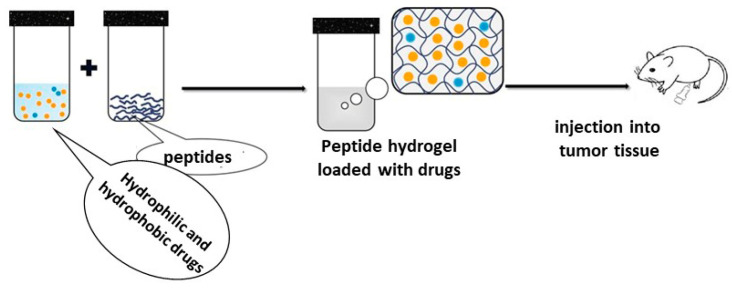
Entrapment of hydrophilic and hydrophobic drugs within peptidic hydrogel for tumor tissue treatments.

**Figure 3 materials-15-05871-f003:**
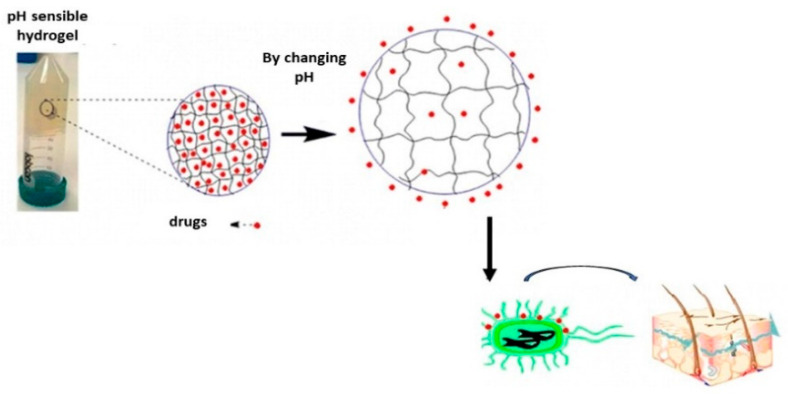
Schematic illustration of hydrogel for antibacterial and wound healing applications.

**Table 1 materials-15-05871-t001:** Some representative examples of physical and chemical methods for the synthesis of peptide-based hydrogels.

Synthesis Method	Hydrogel Composition	Main Interaction	Reference
physical cross-linking	vancomycin-pyrene	hydrogen bonds	[6]
	Fmoc-FFpY	electrostatic interactions	[7]
	Fmoc	hydrophobic interactions	[8]
	*Phg*E*Phg*K (Phg4)	*π*–*π* stacking	[9]
chemical and enzymatic cross-linking	*β*-hairpin peptide	photo-polymerization	[10]
	poly(L-glutamic acid)	enzyme polymerization	[11]
	MAX7CNB	radiation polymerization	[12]

**Table 2 materials-15-05871-t002:** Recent representative examples of biomedical applications of short peptide-based hydrogels.

Classification	Hydrogel Composition	Application	Reference
dipeptides	Boc-*α*-diphenylalanine and Boc-*α*-dityrosine	drug delivery	[19]
	Fmoc-FF with glycol chitosan	drug delivery of doxorubicin (DOX)	[20]
	aromatic di-FF derivative, DFF	drug delivery of 5-aminolevolinic acid	[21]
	aromatic Fmoc-functionalized dipeptides	-	[22]
	Amoc (9-anthracenemethoxycarbonyl)-capped dipeptide	-	[23,24]
	Fmoc-diphenylalanine	biosensing	[25]
	Phe	-	[26]
	dehydrodipeptides	drug delivery of naproxen	[27]
tripeptides	NI-GFF (NI-Gly-L-Phe-L-Phe) and NI-FFG (NI-L-Phe-L-Phe-Gly))	-	[28]
	Fmoc-FFY (Fmoc: 9-fluorenylmethoxycarbonyl; F: phenylalanine; Y: tyrosine)	tissue engineering	[29]
	arginine-glycine-aspartate	tissue engineering	[30]
	Fmoc protected phenylalanine-phenylalanine-cysteine	-	[31]
	Fmoc modified tripeptide hydrogels containing *β*-amino acid	-	[32]
	modified tripeptides (Fmoc-FFF) consisting of Fmoc-phenylalanine amino acid (Fmoc-Phe) and phenylalanine dipeptide (Phe_2_)	-	[33,34,35,36,37,38,39]
tetrapeptides	GDFDFDY	vaccine	[40]
	IVFK and IVZK sequences	-	[41]
	phenolic acid-modified tetrapeptide	Antibiotic drug delivery	[42]
	tetrapeptide hydrogel for encapsulation of protein	vaccine	[43]
	naproxen acid modified tetrapeptide (Npx-GFFY)	-	[44,45]
	Nap-protected tetrapeptide	ophthalmic drug delivery	[46]
pentapeptides	isoleusine and aspartate	-	[47]
	modified pentapeptide	curcumin delivery	[48]
	IKVAV-functionalized poly(lactide ethylene oxide fumarate)	tissue engineering	[49]
	modified pentapeptide	tissue engineering	[50,51]
hexapeptides	a modified hexapeptide	drug delivery of doxorubicin (DOX)	[52]
	amphiphilic cationic peptides		[53]
	a modified hexapeptide	drug delivery of doxorubicin (DOX)	[54,55]
oligopeptides	an injectable octapeptide	drug delivery of doxorubicin (DOX)	[56]
	a heparin-containing oligopeptide	tissue engineering	[57]
	*β*-heparin oligopeptide, Max8	co-delivery of doxorubicin (DOX) and gemcitabine (GEM)	[58]

## Data Availability

Not applicable.

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
