# Peer review of "Peptide-Based Hydrogels: New Materials for Biosensing and Biomedical Applications"

_materials, 2022, doi:10.3390/ma15175871_

Round 1

Reviewer 1 Report

1.     In the introduction, paragraph 1, line 36 “On the other side, they can also be fabricated by chemical and enzymatic cross-linking of peptides (see Table 1).5” What is the meaning of (see Table 1).5? Should the number 5 be a reference?

2.     In the introduction, paragraph 2, line 44 “In peptide hydrogels, amino acids are the building blocks of a 3D matrix and, depending on the number of peptide linkages (–CONH– amide bonds).” In terms of "(–CONH– amide bonds")" should be present as "amide linkage". It will convey a more direct meaning. And all the linkages that are bonded should be presented as the name of the linkage rather than the symbol.

3.     In the introduction, paragraph 6, line 98 “In another study, they employed dipeptides for immobilizing live cells and enzymes as sensing elements into the hydrogel matrix for the detection of superoxide anion (O2•–).” In terms of “(O2•–)” should be present as "oxygen ion". It will convey a more direct meaning. 

4.     In part of biomedical applications of peptide-based hydrogels, an illustration of a biosensing application should be created that shows the structure of the system that can describe how the system could be done.

5.     In part of anticancer drug delivery, an illustration of an anticancer drug delivery should be draw that shows the structure of the system that can explain how peptide-based hydrogels can improve the pharmacokinetic properties of both hydrophilic and hydrophobic drugs should be depicted in a diagram of anticancer drug delivery.

6.     In part of vaccines, line 546, an illustration of vaccines should be details that shows the immune system is critical for the suppression of immune related disorders such as cancer.

7.     In part of vaccines, line 581 to 585, The letters are not the same font and size.

8.     In part of antibacterial and wound healing applications, line 638, an illustration of antibacterial and wound healing applications should be details that shows materials must be created to decrease infections and raise living standards, which will help the social economy and medical resources.

9.     In part of 3D bioprinting and tissue engineering, line 833, an illustration of antibacterial and wound healing applications should be represented that shows the creation of biocompatible materials for the repair or replacement of abnormal tissues or organs is the focus of the interconnected multidisciplinary fields of tissue engineering and 3D bioprinting. Which are employed in tissue engineering, ought to be very friendly with body cells.

Author Response

REVIEWER 1

  1. In the introduction, paragraph 1, line 36 “On the other side, they can also be fabricated by chemical and enzymatic cross-linking of peptides (see Table 1).5” What is the meaning of (see Table 1).5? Should the number 5 be a reference?

A1. We have revised the sentence indicated on line 3.There has been a mistake and the number 5 should be removed. We have modified the text accordingly

  1. In the introduction, paragraph 2, line 44 “In peptide hydrogels, amino acids are the building blocks of a 3D matrix and, depending on the number of peptide linkages (–CONH– amide bonds).” In terms of "(–CONH– amide bonds")" should be present as "amide linkage". It will convey a more direct meaning. And all the linkages that are bonded should be presented as the name of the linkage rather than the symbol.

A2.We thank the reviewer for the suggestion and we agree with the replacement of the term “amide bonds” with a more direct and correct term “amide linkage” in peptide based hydrogels. We have modified the text accordingly

  1. In the introduction, paragraph 6, line 98 “In another study, they employed dipeptides for immobilizing live cells and enzymes as sensing elements into the hydrogel matrix for the detection of superoxide anion (O2•–).” In terms of “(O2•–)” should be present as "oxygen ion". It will convey a more direct meaning.

A3. We thank the reviewer for the suggestion and we agree with the replacement of the term “(O2•–)”with a more direct term "oxygen ion". We have modified the text accordingly.

  1. In part of biomedical applications of peptide-based hydrogels, an illustration of a biosensing application should be created that shows the structure of the system that can describe how the system could be done.

A4.We agree with the reviewer suggestion and we included in the text an illustration on hydrogels application for biosensing.

  1. In part of anticancer drug delivery, an illustration of an anticancer drug delivery should be draw that shows the structure of the system that can explain how peptide-based hydrogels can improve the pharmacokinetic properties of both hydrophilic and hydrophobic drugs should be depicted in a diagram of anticancer drug delivery.

A5.We agree with the reviewer suggestion and we draw a very simple illustration on hydrogel drug delivery ability by injection in animal tissues and we inserted it in the text accordingly

  1. In part of vaccines, line 546, an illustration of vaccines should be details that shows the immune system is critical for the suppression of immune related disorders such as cancer.

A6. We thank the reviewer for the suggestion however we think that this specific figure on vaccines is unnecessary and that the added bibliographic references are able to explain better the content

  1. In part of vaccines, line 581 to 585, The letters are not the same font and size.

A7. We thank the reviewer for pointing out the error. We have revised the text in the mentioned paragraph suggested accordingly.

  1. In part of antibacterial and wound healing applications, line 638, an illustration of antibacterial and wound healing applications should be details that shows materials must be created to decrease infections and raise living standards, which will help the social economy and medical resources.

A8. We agree with the reviewer suggestion and we draw an illustration on hydrogel application in wound healing. We have inserted in the text the illustration above described accordingly

  1. In part of 3D bioprinting and tissue engineering, line 833, an illustration of antibacterial and wound healing applications should be represented that shows the creation of biocompatible materials for the repair or replacement of abnormal tissues or organs is the focus of the interconnected multidisciplinary fields of tissue engineering and 3D bioprinting. Which are employed in tissue engineering, ought to be very friendly with body cells.

A9. We thanks the reviewer for the suggestion but we think that the image explaining the topics mentioned in the paragraph is unnecessary.

Reviewer 2 Report

The present review is well-written with great deal of attention to details.  Until now, reviews dedicated to this topic focused more on the mechanism of gelation, type of functionalization, methods of synthesis, etc. In this work all the data is organized based on the application of the peptide hydrogels, which might be useful for the readers in some cases. 

However, for me pesonally it was a little hard to perceive the data like this. I do not insist on making changes to the text, but I suggest that the authors think about it. For example, in line 106 it says "Tripeptides are the next class of short peptides which can be N-terminal protected 106 [29] or non-protected [13]."  First of all, tripeptides are not the only peptides that can be protected or non-protected. Secondly, it is not discussed, how being protected or non-protected affects the properties of the hydrogels. And this is a common theme: I lacked the analysis of structure-property relationship. 

I would recommend to put the reference to Table 2 somwhere else, because it reads as if the table is supposed to prove that short-chain peptides are "cost effective and have easier syn-46 thesis methods compared to long chain ones", and it does not. 

Finally, I do not understand, why some of the previous reviews and works on the topic (such as doi.org/10.1039/C9SM02127B or /10.1039/C3RA40234G or some others)were not cited.  It might be helpful to direct the reader to other types of peptide hydrogel classifications. 

The manuscript is comprehensive and the text only requires a minor typo- and spell-check. Here are a few lines, in which I was able to find either a typo or a spelling mistake: 36, 38, 92, 132, 195, 199, 201, 202, 213, 215, 246. There are obviously more that need to be found. 

Overall, I am very happy to see such a nice review and I advise to accept it after a minor revision. Besides, it fits the topic of the special issue perfectly, which is always great. 

I wish all the best to the authors!

Author Response

REVIEWER 2

The present review is well-written with great deal of attention to details.  Until now, reviews dedicated to this topic focused more on the mechanism of gelation, type of functionalization, methods of synthesis, etc. In this work all the data is organized based on the application of the peptide hydrogels, which might be useful for the readers in some cases.

However, for me pesonally it was a little hard to perceive the data like this. I do not insist on making changes to the text, but I suggest that the authors think about it. For example, in line 106 it says "Tripeptides are the next class of short peptides which can be N-terminal protected 106 [29] or non-protected [13]."  First of all, tripeptides are not the only peptides that can be protected or non-protected. Secondly, it is not discussed, how being protected or non-protected affects the properties of the hydrogels. And this is a common theme: I lacked the analysis of structure-property relationship.

We agree with the reviewer suggestion and we think that, as suggested, many peptides may be N-terminal protected. The chemical reactions involved in peptides N-terminal protection affect, obviously,  modify their structure and also final physico-chemical properties.

I would recommend to put the reference to Table 2 somewhere else, because it reads as if the table is supposed to prove that short-chain peptides are "cost effective and have easier syn-46 thesis methods compared to long chain ones", and it does not.

We thank the reviewer for the suggestion and we have moved the reference to table 2 below(line 60) in a more specific paragraph,  accordingly

Finally, I do not understand, why some of the previous reviews and works on the topic (such as doi.org/10.1039/C9SM02127B or /10.1039/C3RA40234G or some others)were not cited.  It might be helpful to direct the reader to other types of peptide hydrogel classifications.

We thank the reviewer for the suggestion. We have added in the text the cited references accordingly

The manuscript is comprehensive and the text only requires a minor typo- and spell-check. Here are a few lines, in which I was able to find either a typo or a spelling mistake: 36, 38, 92, 132, 195, 199, 201, 202, 213, 215, 246. There are obviously more that need to be found.

We revised all the text for typo and spell-errors as suggested and we have modified it accordingly.

Overall, I am very happy to see such a nice review and I advise to accept it after a minor revision. Besides, it fits the topic of the special issue perfectly, which is always great.

I wish all the best to the authors!